# Heterogeneity in Multi-Agent Reinforcement Learning

## Abstract

*Heterogeneity* is a fundamental property in multi-agent reinforcement learning (MARL), which is closely related not only to the functional differences of agents, but also to policy diversity and environmental interactions. However, the MARL field currently lacks a rigorous definition and deeper understanding of heterogeneity. This paper systematically discusses heterogeneity in MARL from the perspectives of *definition*, *quantification*, and *utilization*. First, based on an agent-level modeling of MARL, we categorize heterogeneity into five types and provide mathematical definitions. Second, we define the concept of heterogeneity distance and propose a practical quantification method. Third, we design a heterogeneity-based multi-agent dynamic parameter sharing algorithm as an example of the application of our methodology. Case studies demonstrate that our method can effectively identify and quantify various types of agent heterogeneity. Experimental results show that the proposed algorithm, compared to other parameter sharing baselines, has better interpretability and stronger adaptability. The proposed methodology will help the MARL community gain a more comprehensive and profound understanding of heterogeneity, and further promote the development of practical algorithms.

## 1 Introduction

Multi-agent reinforcement learning (MARL) has achieved success in various real-world applications, such as swarm robotic control [Kalashnikov et al., 2018], autonomous driving [Zhou et al., 2021], and large language model fine-tuning [Ma et al., 2024]. However, most MARL studies focus on policy learning for homogeneous multi-agent systems (MAS), overlooking in-depth discussions of heterogeneous multi-agent scenarios [Ning and Xie, 2024]. *Heterogeneity* is a common phenomenon in multi-agent systems. For example, in nature, different species of fish collaborate to find food [Burns et al., 2019]; in human society, diverse teams demonstrate higher intelligence and resilience [Dall'Anese et al., 2013, Young, 1993]; and in artificial systems, aerial drones and ground vehicles cooperate to monitor forest fires [Lwowski et al., 2017]. Heterogeneity can enhance system functionality, reduce costs, and improve robustness, but effectively leveraging heterogeneity remains a key challenge in multi-agent system [Bennett, 2024]. As an approach of learning through environmental interactions, MARL can effectively enable multi-agent systems to learn collaborative policies. Hence, exploring heterogeneity from a reinforcement learning perspective would significantly broaden the applicability of MARL.

In the current MARL field, although some works explicitly or implicitly mention agent heterogeneity, only a few focus on its definition and identification. Regarding explicit discussion of heterogeneity, studies have explored communication issues [Seraj et al., 2021], credit assignment [Yu et al., 2024], and zero-shot generalization [Guo et al., 2024] in heterogeneous MARL. However, these works limit their focus to agents with clear functional differences and lack definitions of agent heterogeneity. On the other hand, many studies explore policy diversity in MARL. Some encourage agents to learn distinguishable behaviors based on identity or trajec-

tory information [Jiang and Lu, 2021, Li et al., 2021], some works group agents using specific metrics [Wang et al., 2021, Christianos et al., 2021], and some quantify policy differences [Bettini et al., 2023b, Hu et al., 2024] and design algorithms to control policy diversity [Bettini et al., 2024].

However, these works do not adequately address where policy diversity originates or how it fundamentally relates to agent differences. In terms of defining and classifying heterogeneity in MARL, [Bettini et al., 2023a] divides heterogeneity into physical and behavioral types but lacks a mathematical definition. [Seraj et al., 2021] provides extended POMDP for heterogeneous MARL settings, but do not classify or define heterogeneity. Others introduce the concept of local transition heterogeneity [Yu et al., 2024], but does not cover all elements of MARL. Overall, heterogeneity is not only a characteristic that exists in MAS with traditional functional differences, but also a fundamental property across the entire MARL field. Currently, there is still a lack of *systematic analysis of agent heterogeneity from the MARL perspective*.

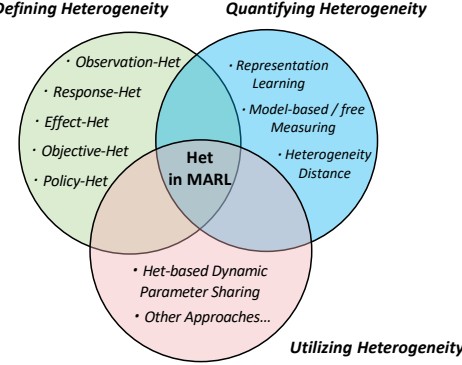

Figure 1: Our Philosophy. We aim to systematically discuss heterogeneity in MARL, establishing methodologies for defining, quantifying and utilizing heterogeneity.

To fill the aforementioned gaps, we conduct a series of studies on defining, quantifying, and utilizing heterogeneity from the perspective of MARL, the philosophy of our study can be found in Figure 1. Our contributions are summarized as follows:

• **Defining Heterogeneity:** Based on an agent-level model of MARL, we categorize heterogeneity into observation heterogeneity, response transition heterogeneity, effect transition heterogeneity, objective heterogeneity, and policy heterogeneity, and provide corresponding definitions.

• **Quantifying Heterogeneity:** We define the heterogeneity distance, and propose a quantification method based on representation learning, applicable to both model-free and model-based settings. Additionally, we give the concept of meta-transition heterogeneity to quantify agents' comprehensive heterogeneity.

• **Utilizing Heterogeneity:** We develop a multi-agent dynamic parameter-sharing algorithm based on heterogeneity quantification, which offers better interpretability and fewer task-specific hyperparameters compared to other related parameter-sharing algorithms.

In this paper, we adopt a discussion approach that progresses *from theory to practice* and *from general to specific*. The overall structure is organized as follows: Section 2 introduces the agent-level modeling of the MARL primal problem; Section 3 provides the classification and definition of heterogeneity in MARL; Section 4 proposes the method for quantifying heterogeneity and presents case studies; Section 5 describes the dynamic parameter-sharing algorithm; Section 6 provides the related experimental results; and Section 7 summarizes the entire paper.

## 2 Preliminaries

**Primal Problem of MARL.** In this paper, we use Partially Observable Markov Game (POMG) [Littman, 1994, Kochenderfer et al., 2022] as the general model for the primal problem of MARL.[1] To better study agent heterogeneity, we adopt an agent-level modeling approach similar to that in [Seraj et al., 2021, Gronauer and Diepold, 2022]. A POMG is defined as an 8-tuple, represented as follows:

$$\text{POMG} := \langle N, \{S^i\}_{i \in N}, \{O^i\}_{i \in N}, \{A^i\}_{i \in N}, \{\Omega^i\}_{i \in N}, \{\mathcal{T}^i\}_{i \in N}, \{r_i\}_{i \in N}, \gamma \rangle, \tag{1}$$

Among all elements in equation 1, $N$ is the set of all agents, $\{S^i\}_{i \in N}$ is the global state space which can be factored as $\{S^i\}_{i \in N} = \times_{i \in N} S^i \times S^E$, where $S^i$ is the state space of an agent $i$, and $S^E$ is the environmental state space, corresponding to all the non-agent components. $\{O^i\}_{i \in N} = \times_{i \in N} O^i$ is

---

[1]POMG is an extension of POMDP for multi-agent settings, with the basic extension path being MDP → POMDP → POMG [Sun et al., 2023]. Please refer to Appendix C to see a more detailed explanation of POMG.

the joint observation space and $\{A^i\}_{i \in N} = \times_{i \in N} A^i$ is the joint action space of all agents. $\{\Omega^i\}_{i \in N}$ is the set of observation functions. $\{\mathcal{T}^i\}_{i \in N} = (\mathcal{T}^1, \cdots, \mathcal{T}^{|N|}, \mathcal{T}^E)$ is the collection of all agents' transitions and the environmental transition. Finally, $\{r_i\}_{i \in N}$ is the set of reward functions of all agents and $\gamma$ is the discount factor.

Here, we give the independent and dependent variables for each function and their notation. At each time step $t$, an agent $i$ receives an observation $o_t^i \sim \Omega^i(\cdot|\hat{s}_t)$, where $\hat{s}_t \in \{S^i\}_{i \in N}$ is the global state at time $t$. Then, agent $i$ makes a decision based on its observation, resulting in an action $a_t^i \sim \pi_i(\cdot|o_t^i)$. The environment then collects actions from all agents to form the global action $\hat{a}_t = (a_t^1, \ldots, a_t^{|N|})$. We assume that the local state transition of agent $i$ is influenced by the global state and global action, so its local state transitions to a new state $s_{t+1}^i \sim \mathcal{T}^i(\cdot|\hat{s}_t, \hat{a}_t)$. Similarly, the states of other agents and the environment also transition, yielding the next global state $\hat{s}_{t+1} = (s_{t+1}^1, \ldots, s_{t+1}^{|N|}, s_{t+1}^E) \sim (\mathcal{T}^1(\cdot|\hat{s}_t, \hat{a}_t), \ldots, \mathcal{T}^{|N|}(\cdot|\hat{s}_t, \hat{a}_t), \mathcal{T}^E(\cdot|\hat{s}_t, \hat{a}_t)) = \{\mathcal{T}^i\}_{i \in N}(\cdot|\hat{s}_t, \hat{a}_t)$. At the same time, all agents receive rewards, with the reward for a specific agent $i$ given by $r_t^i \sim r^i(\cdot|\hat{s}_t, \hat{a}_t)$.

The objective of MARL is to solve POMG by finding an optimal joint policy that maximizes the cumulative reward for all agents. We denote the individual optimal policy for agent $i$ as $\pi_i^*$ and the optimal joint policy as $\hat{\pi}^*$, which can be expressed as $\hat{\pi}^* = (\pi_1^*, \ldots, \pi_{|N|}^*)$. The optimal joint policy for a POMG can be obtained through the following equation:

$$\pi_i^* = \arg\max_{\hat{\pi}} \mathbb{E}_{\hat{\pi}} \left[ \sum_{k=0}^{\infty} \gamma^k \sum_{i \in N} r_{t+k}^i \Big| \hat{s}_t = \hat{s}_0 \right], \tag{2}$$

where $\gamma$ is the discount factor, and the expectation is taken over the trajectories induced by the joint policy $\hat{\pi}$ starting from the initial global state $\hat{s}_0$.

# 3 Definition and Taxonomy of Heterogeneity in MARL

**Heterogeneity in MAS.** Our goal is to define agent heterogeneity from the perspective of MARL. Before achieving this, we need to discuss heterogeneity in MAS across various disciplines. Early studies [Dudek et al., 1996, Parker, 2000] define heterogeneity as differences in physical structure or functionality of agents, which aligns with common understanding. Later work [Panait and Luke, 2005] describes heterogeneity as differences in agent behavior, further expanding its meaning. Recently, [Bennett, 2024] points out that heterogeneity may be a complex phenomenon, related not only to the inherent properties of agents, but also to their interactions with environment. Thus, heterogeneity in MARL should not be limited to inherent functional differences of agents, but should also fully consider various coupling effects of agents within the environment.

**Heterogeneity in MARL.** In the context of MARL, the fundamental modeling of MARL and its primal problem provides considerable convenience for defining heterogeneity. This modeling clearly specifies all MARL elements, delineating the boundaries of the problem discussion [2] and ensuring the completeness of the discussion.

We focus on the heterogeneity *among agents* within a same POMG. As discussed in Section 2, the function in a POMG can connect agent-level elements. Therefore, we categorize agent heterogeneity into five types centered around the functions. This approach not only avoids overly redundant classification but also ensures comprehensive coverage of each agent-level element. Regarding definition, the condition for heterogeneity is obtained by *taking the negation of the necessary and sufficient conditions for homogeneity*.

Specifically, these five types of heterogeneity and their related definitions are as follows:

• *Observation heterogeneity* describes the differences of agents in observing global information. The relevant elements include the agent's observation space and observation function.

**Definition 1.** Agents $i$ and $j$ are observation heterogeneous if the following conditions do not hold at the same time: ① $O^i = O^j$; ② $\forall \hat{s} \in \{S^i\}_{i \in N}, \Omega^i(\cdot|\hat{s}) = \Omega^j(\cdot|\hat{s})$.

---

[2]In this paper, we focus on the heterogeneity of MARL under the conventional POMG problem. Additional discussions on unconventional heterogeneity types are provided in Appendix D.

- *Response transition heterogeneity* describes the differences of agents in how their state transitions are affected by global environment components (*environment-to-self*). The relevant elements include the agent's state space and local state transition function.

**Definition 2.** Agents $i$ and $j$ are response transition heterogeneous if the following conditions do not hold at the same time: ① $S^i = S^j$; ② $\forall \hat{s} \in \{S^i\}_{i \in N}, \hat{a} \in \{A^i\}_{i \in N}, \mathcal{T}^i(\cdot|\hat{s}, \hat{a}) = \mathcal{T}^j(\cdot|\hat{s}, \hat{a})$.

- *Effect transition heterogeneity* describes the differences of agents in how their states and actions impact global state transitions (*self-to-environment*). The relevant elements include the agent's action space, state space, and global state transition function.

**Definition 3.** Agents $i$ and $j$ are effect transition heterogeneous if the following conditions do not hold at the same time: ① $S^i = S^j$; ② $A^i = A^j$; ③ $\forall s' \in S^{-i}, a' \in A^{-i}, s \in S^i, a \in A^i, \mathcal{T}^{-i}(\cdot|s', s, a', a) = \mathcal{T}^{-j}(\cdot|s', s, a', a)$.

In the above definition, $S^{-i} = \times_{k \in N, k \neq i} S^k \times S^E$ represents the joint state space of all agents except agent $i$, reflecting the influence of the agent on other states. Similarly, $A^{-i}$ denotes the joint action space excluding agent $i$, and $\mathcal{T}^{-i}$ is the collection of state transitions excluding agent $i$.

- *Objective heterogeneity* describes the differences of agents in the objective they aim to achieve. The relevant element is the agent's reward function.

**Definition 4.** Agents $i$ and $j$ are objective heterogeneous if the following condition do not hold:

① $\forall \hat{s} \in \{S^i\}_{i \in N}, \hat{a} \in \{A^i\}_{i \in N}, r^i(\cdot|\hat{s}, \hat{a}) = r^j(\cdot|\hat{s}, \hat{a})$.

- *Policy heterogeneity* describes the differences of agents in their autonomous decision-making based on observations. The relevant elements include the observation space, action space, and policy.

**Definition 5.** Agents $i$ and $j$ are policy heterogeneous if the following conditions do not hold at the same time: ① $O^i = O^j$; ② $A^i = A^j$; ③ $\forall o \in O^i, \pi_i(\cdot|o) = \pi_j(\cdot|o)$.

In the five types of heterogeneity mentioned above, we assume that all functions follow the Markov property, making them independent of the agent's trajectory. Therefore, the first four types of heterogeneity can be considered environment-related, which reflect the heterogeneity in the MARL primal problem. The last type describes the policy heterogeneity of agents before, during, and after training, which reflects the heterogeneity of optimization objectives (policies) in the primal problem.

# 4 Quantifying Heterogeneity in MARL

## 4.1 Heterogeneity Distance Based on Representation Learning

**Heterogeneity Distance.** In this section, we present the method to quantify the above five types of heterogeneity. According to the definition, each type of heterogeneity corresponds to a core function which connects relevant elements in the heterogeneity type. Therefore, we quantify the differences in these core functions to characterize the degree of heterogeneity.[3] To make the quantification results simpler and more practical, we propose the concept of heterogeneity distance.

Let the core function corresponding to a certain heterogeneity type $F$ be denoted as $y \sim F(\cdot|x)$. The formula for calculating the $F$-heterogeneous distance between two agents $i$ and $j$ is given by:

$$d_{ij}^F = \int_{x \in X} D[F_i(\cdot|x) \| F_j(\cdot|x)] \cdot p(x)\, dx, \tag{3}$$

where $X$ is the space of independent variables, $p(x)$ is the probability density function, and $D[\cdot \| \cdot]$ is a measure that quantifies the difference between distributions. The core idea of heterogeneity distance is to examine the cumulative differences between two agents' functions throughout the space of independent variables, which captures any potential local differences. When the independent variables $x$ consist of multiple factors, the above integral becomes a multivariate integral. Based on Equation 3, we provide the specific expressions for quantifying all heterogeneous distances in Appendix F and discuss the properties of heterogeneous distance below.

---

[3]Quantifying space elements is feasible and even easier to implement. But a space element may appear across multiple heterogeneity types, making it unsuitable as unique identifiers for specific heterogeneity types.

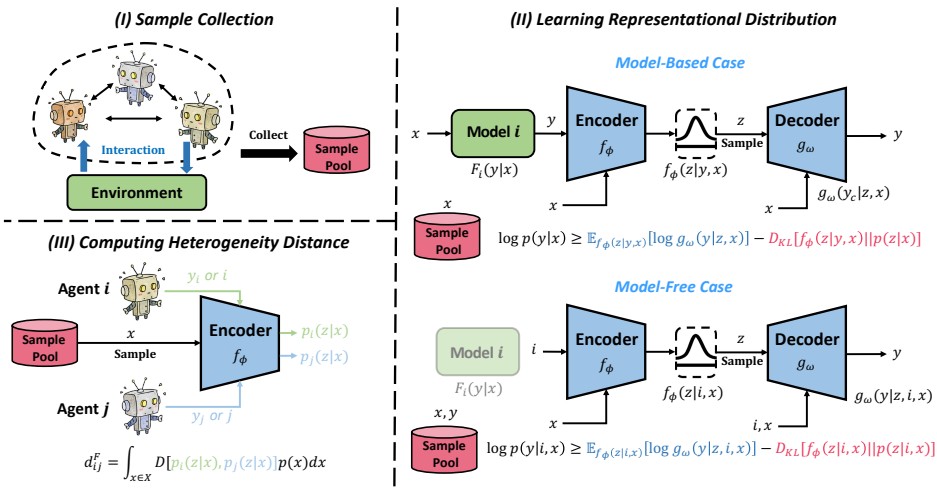

Figure 2: The method of measuring heterogeneity distance based on representation learning.

**Proposition 1**. (*Properties of Heterogeneity Distance*) ① *Symmetry*: $d_{ij}^F = d_{ji}^F$; ② *Non-negativity*: $d_{ij}^F \geq 0$; ③ *Identity of indiscernibles*: $d_{ij}^F = 0$ if and only if agents $i$ and $j$ are $F$-homogeneous; ④ *Triangle inequality*: $d_{ij}^F \leq d_{ik}^F + d_{kj}^F$ $(i, j, k \in N)$. This proposition holds as long as the measure $D$ satisfies ①②③④. The proof is provided in Appendix E.

**Practical Method.** Although the heterogeneity distance has a simple form, some issues may arise during practical computation. First, computing the distribution distance via sampling is computationally complex, while computing the distance using analytical solutions requires knowing the distribution type. In real-world scenarios, the distributions may be unknown or of different types [4]. Second, the independent variable space may be very large, making traversal-based computation infeasible.

For the first issue, our approach is to **standardize the original distributions**. By learning a representation mapping, for all independent variables $x$, a measurable distribution $p_i(z|x)$ is used to capture the characteristics of the original distribution $F_i(y|x)$, replacing the original one for measure computation. For the second issue, our approach is **sampling based on the interaction between agents and the environment**. Instead of simply traversing the space or using random policy exploration for sampling, we construct a sample pool using trajectories from the training phase of MARL. This significantly reduces computational load and filters out excessive marginal spaces that interfere with MARL, benefiting the use of heterogeneity distance in subsequent MARL tasks (Section 5). Combining these ideas, we propose a practical method as shown in Figure 2.

**In the first step**, the agents interact with the environment during MARL training to build a sample pool. Notably, the sample pool data is shuffled to ensure that the learned function follows the Markov property (independent of historical information), similar to the original function.

**In the second step**, the representational distributions are learned. We discuss this in both model-based and model-free settings, corresponding to cases where the function is known and unknown, respectively. We adopt the conditional variational autoencoder (CVAE) framework [Sohn et al., 2015] for representation learning. In the model-based case, CVAE performs a reconstruction task [Lopez-Martin et al., 2017]. The optimization goal is to maximize the likelihood of the reconstructed variable $\log p(y|x)$. Through derivation, we obtain the evidence lower bound (ELBO) as:

$$ELBO_{\text{model-based}} = \mathbb{E}_{f_\phi(z|y,x)} \left[ \log g_\omega(y|z,x) \right] - D_{KL} \left[ f_\phi(z|y,x) \parallel p(z|x) \right], \quad (4)$$

where $f_\phi$ and $g_\omega$ represent the encoder and decoder, respectively, and $p(z|x)$ is the prior conditional latent distribution. We designed the relevant loss based on ELBO, including a reconstruction term and a prior-matching term. The derivation for this part can be found in Appendix H.

---

[4]For example, the action distribution of an agent $i$ is a Gaussian distribution, while that of agent $j$ is a bimodal distribution.

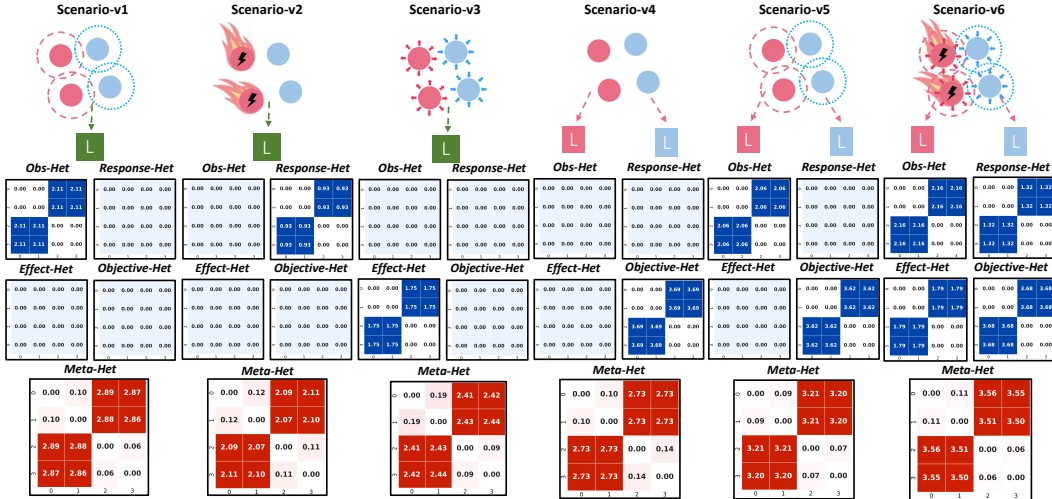

Figure 3: The scenario illustration and heterogeneity distance matrices in our case study. In v1, the observations of agents in different groups are shuffled in different orders. In v2, the max move speed of agents in different groups is different. In v3, one group of agents applies repulsive force to surrounding entities, while the other applies attractive force. In v4, agents in different groups need to move to different landmarks. In v5, both the observations and objectives of agents are heterogeneous. In v6, all the above properties of agents are heterogeneous. Below each scenario illustration, the corresponding heterogeneity distance matrices are shown. Specifically, *Obs-Het*, *Response-Het*, *Effect-Het*, *Objective-Het*, and *Meta-Het* correspond to observation / response transition / effect transition / objective / meta-transition heterogeneity, respectively.

In the model-free case, CVAE essentially performs a prediction task [Zhang et al., 2021], capturing the model characteristics of each agent. The optimization goal is to maximize the likelihood of the predicted variable $y$ given conditions $i$ and $x$, where $i$ is the agent ID. Similarly, we derive the corresponding ELBO:

$$ELBO_{\text{model-free}} = \mathbb{E}_{f_\phi(z|i,x)} \left[ \log g_\omega(y|z,i,x) \right] - D_{KL} \left[ f_\phi(z|i,x) \parallel p(z|i,x) \right]. \tag{5}$$

**In the third step**, the heterogeneity distances for multi-agents are computed. For each $x$, we obtain the distribution representation using the encoder in either the model-based or model-free manner. The distance under a specific $x$ is computed using the *Wasserstein distance* [Vaserstein, 1969] of the prior distribution (*standard Gaussian*). The heterogeneity distance is then calculated via multi-rollout Monte Carlo sampling. In practice, we parallelize this operation [5], enabling simultaneous computation of distances between all agents on GPUs, significantly improving computational efficiency.

**Meta-Transition.** The aforementioned method can quantify the heterogeneity of agents for specific types. In practical applications, researchers may also want to quantify the **comprehensive** heterogeneity of agents to enable operations such as grouping. To this end, we give the *Meta-Transition* model (see Appendix G for details). By measuring the differences between meta-transitions, the comprehensive heterogeneity related to environment can be quantified. We refer to this as the meta-transition heterogeneity distance.

## 4.2 Case Study

We design a multi-agent spread scenario for case study. In the basic scenario, there are two groups, each with two agents, and their goal is to move close to randomly generated landmarks. Based on the basic scenario, we create 6 extended versions to show the quantitative results of different types of heterogeneity and meta-transition heterogeneity. As shown in Figure 3, the first 4 versions correspond to the 4 environment-related types of heterogeneity, while the last 2 versions represent

---

[5]Our code is provided in the supplementary material.

227 cases where multiple types of heterogeneity exist. We use the model-based manner to compute the
228 four heterogeneity distance matrices mentioned above, and the model-free manner to compute the
229 meta-heterogeneity distance matrix for the agents.

230 The results show that for each type of heterogeneity, our method can accurately capture and identify
231 the differences. For meta-transition heterogeneity, the distance between agents in the same group is
232 much smaller than that in different groups. Moreover, as the number of heterogeneity types increases,
233 the distance between different groups also increases. These results demonstrate the effectiveness of
234 our method for various environment-related heterogeneities.

235 We further quantify the policy heterogeneity dis-
236 tance (*Policy-Het*) and meta-transition hetero-
237 geneity distance (*Meta-Het*) of agents during the
238 training process. We select two algorithms at the
239 extreme ends of parameter sharing: fully parame-
240 ter sharing (FPS) and no parameter sharing (NPS)
241 for training in the above scenarios. Figure 4
242 shows the measurement results at 500 and 1500
243 updates. From the *Policy-Het* results, the policy
244 distance can effectively reveal the evolutionary
245 relationship of agent policy differences in MARL.
246 From the *Meta-Het* results, the comprehensive
247 agent heterogeneity measurement remains con-
248 sistent across different learning algorithms, and
249 can identify environmental heterogeneous char-
250 acteristics in scenarios more rapidly compared to
251 policy evolution.

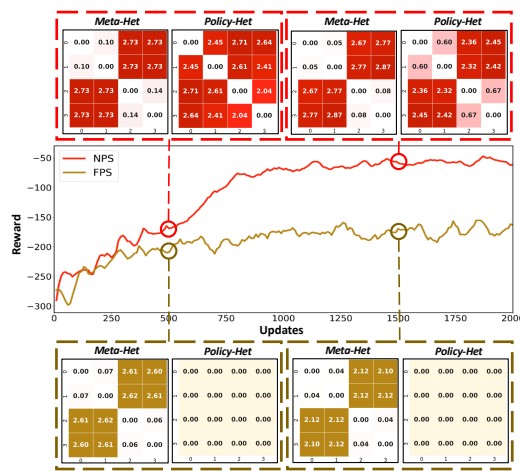

Figure 4: Meta-transition heterogeneity and pol-
icy heterogeneity distance matrices during train-
ing in our case study.

## 5 Multi-Agent Dynamic Parameter Sharing Based on Heterogeneity Quantification: An Application

255 Based on the case study in Section 4.2, the pro-
256 posed method can not only accurately quantify all
257 types of heterogeneity, but also the comprehen-
258 sive heterogeneity among agents. Additionally, the method is independent of the parameter-sharing
259 type used in MARL and can be deployed online, thereby further enhancing its practicality. In
260 this section, we provide a practical application of our methodology to demonstrate its potential in
261 empowering MARL.

262 We select parameter sharing in MARL as our application context. As a common technique in
263 MARL, parameter sharing can reduce computational consumption while improving sample utilization
264 efficiency [KIM and Sung, 2023], but its excessive use may inhibit agents' policy heterogeneity
265 expression [Hu et al., 2024]. Many works have attempted to find a balance between parameter sharing
266 and policy heterogeneity [Li et al., 2024b]. However, existing approaches suffer from two main
267 problems: *poor interpretability*, unable to explain why policy heterogeneity is necessary and to what
268 extent; and *poor adaptability*, manifested by numerous task-specific hyperparameters and inability
269 to dynamically adapt policy training. (For a more detailed discussion of these algorithms, see the
270 experimental section 6.1)

271 To address these issues, we propose a Heterogeneity-based multi-agent Dynamic Parameter Sharing
272 algorithm (HetDPS) with two core ideas(More details can be found in Appendix I):

273 ♠ **Grouping agents for parameter sharing through heterogeneity distances**. We utilize distance-
274 based clustering methods to group agents, thus avoiding the introduction of task-specific hyperpa-
275 rameters like group number [Christianos et al., 2021, Li et al., 2024a] or fusion thresholds [Hu et al.,
276 2024]. The heterogeneity distance matrices also enhance the algorithm's interpretability.

277 ♣ **Periodically quantifying heterogeneity and modifying agents' parameter sharing paradigm**.
278 This can help the sample pool become more aligned with policy training. This approach can also
279 help policies escape local optima [Lyle et al., 2024], the effectiveness of such a mechanism has been

verified in the MARL domain [Li et al., 2024b], and even in broader RL areas such as large model fine-tuning [Noukhovitch et al., 2023, Ma et al., 2024].

# 6 Experiments

In the experimental section, we conduct comprehensive comparisons between HetDPS and other parameter sharing algorithms. Beyond performance comparisons, we also analyze the heterogeneity characteristics of each MARL task with our proposed methodology, to demonstrate the algorithm's interpretability. Additionally, we conduct hyperparameter experiments and efficiency and resource consumption experiments to show the adaptability and practicality of HetDPS.

## 6.1 Experimental Setups

**Environments. Partical-based Multi-agent Spreading** [Hu et al., 2024] is a typical environment in the policy diversity domain. In this environment, multiple agents are randomly generated in the center of the map, while multiple landmarks are randomly generated near the periphery. Both agents and landmarks have various colors, and agents need to move to landmarks with matching colors. Additionally, agents need to form tight formations when

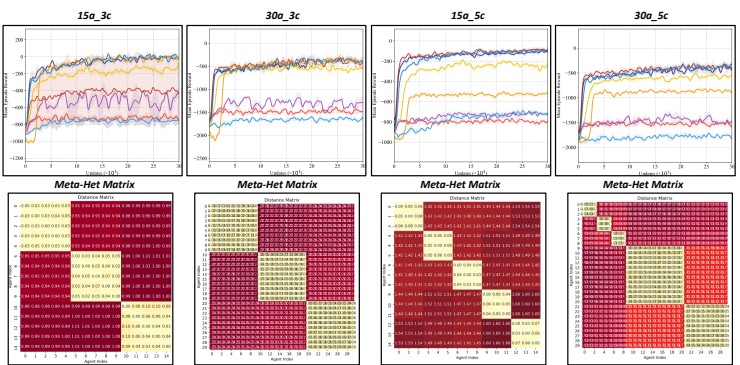

Figure 5: Results on Partical-based Multi-agent Spreading.

they reach the vicinity of landmarks. We employ 4 typical tasks, corresponding to different numbers and color distributions, as detailed in Table 1. **The StarCraft Multi-Agent Challenge (SMAC)** [Samvelyan et al., 2019] is a popular MARL benchmark, where multiple ally units controlled by MARL algorithms aim to defeat enemy units controlled by the game's built-in AI.

**Baselines and training.** We compare HetDPS with other parameter sharing baselines, as listed in Table 2. We analyze these baselines along three dimensions: parameter sharing paradigm, adaptability, and relationship with heterogeneity utilization. As seen from the table, current methods can not effectively utilize heterogeneity. Although some methods implicitly use certain heterogeneity quantification results, the elements they involve are not comprehensive. MADPS, as the only method that explicitly uses policy distance for dynamic grouping, relies on the assumption that policy learning can effectively capture

Table 1: Task information for particle-based multi-agent spreading.

| Task | Agent Type Distribution |
| --- | --- |
| *15a_3c* | $5 - 5 - 5$ |
| *30a_3c* | $10 - 10 - 10$ |
| *15a_5c* | $3 - 3 - 3 - 3 - 3$ |
| *30a_5c* | $3 - 3 - 3 - 12 - 9$ |

heterogeneity, which lacks practicality. We use official implementations of the baselines where available. For further discussion on related work and experiments in this paper, see the supplementary materials.

## 6.2 Results

**Performance and interpretability.** We tested the performance of all comparison algorithms in the two environments mentioned above. The reward curves and corresponding heterogeneity distance matrices are shown in Figure 5 and Figure 6. From the reward curve results, we can see that HetDPS achieves either optimal or comparable results in all tasks above.

We quantified the meta-transition heterogeneity distances for all tasks. The results show that our heterogeneity quantification results in the Multi-agent Spreading scenario are highly consistent with

Table 2: Comparison of different methods and their properties.

| Method | Paradigm | Adaptive | Relation to Heterogeneity Utilization |
|---|---|---|---|
| NPS | No Sharing | No | None |
| FPS | Full Sharing | No | None |
| FPS+id | Full Sharing | No | None |
| Kaleidoscope [Li et al., 2024b] | Partial Sharing | Yes | No utilization, increases agent policy heterogeneity as the bias |
| SePS [Christianos et al., 2021] | Group Sharing | No | Implicitly utilizes objective heterogeneity and response transition heterogeneity |
| AdaPS [Li et al., 2024a] | Group Sharing | Yes | Implicitly utilizes objective heterogeneity and response transition heterogeneity |
| MADPS [Hu et al., 2024] | Group Sharing | Yes | Explicitly utilizes policy heterogeneity only |
| HetDPS (ours) | Group Sharing | Yes | Explicitly utilizes heterogeneity, leveraging heterogeneous distance |

Table 3: Training efficiency metrics across different methods. Results are normalized with respect to the FPS method, and averaged across all tasks.

| | NPS | FPS | FPS+id | Kaleidoscope | SePS | AdaPS | MADPS | HetDPS (ours) |
|---|---|---|---|---|---|---|---|---|
| **Training Speed** | 0.952x | 1.000x | 0.992x | 0.974x | 0.986x | 0.614x | 0.539x | 0.712x |

the type distribution in Table 1. This demonstrates the effectiveness of our method in identifying agent heterogeneity. Additionally, we made some interesting discoveries in the SMAC environment. We found that in simpler tasks like *3s5z* and *MMM*, the agent heterogeneity quantification results often do not closely match the original agent types. In *MMM*, agents even tend toward homogeneous policies to improve training efficiency. However, in more difficult tasks such as *3s5z_vs_3s6z* and *MMM2*, agents' quantification results closely match their original types to achieve better coordination. This confirms our view that agent heterogeneity is related not only to the agents' original functional attributes but also to how agents interact with the environment.

**Cost Analysis.** We conducted an experiment to investigate training efficiency. The experimental results are shown in Table 3. The results indicate that although our method introduces periodic heterogeneity quantification, it does not significantly reduce algorithm efficiency.

# 7 Conclusion

Heterogeneity manifests in various aspects of MARL. It is not only related to the inherent properties of agents themselves but also to the coupling factors arising from agent-environment interactions. Consequently, agents that appear homogeneous may develop heterogeneity under environmental influences. In this paper, we categorize heterogeneity in MARL into five types and provide respective definitions. Meanwhile, we propose methods for quantifying

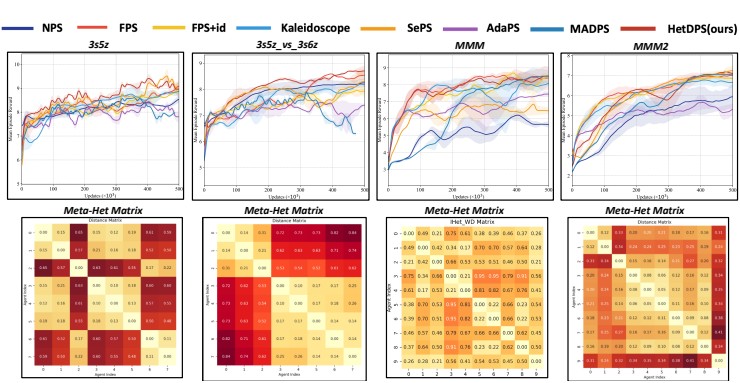

Figure 6: Results on StarCraft Multi-Agent Challenge.

these heterogeneity types and conduct case studies. Under our theoretical framework, policy diversity is merely a manifestation of policy heterogeneity, fundamentally originating from the division of labor necessitated by agents' environmental heterogeneity (*cause*), serving as an inductive bias (*result*) for solving optimal joint policies. Thus, we introduce the quantification of heterogeneity as prior knowledge into multi-agent parameter-sharing learning. The result is HetDPS, an algorithm with strong interpretability and adaptability. HetDPS is not the endpoint of our research, but rather a starting point for heterogeneity applications. We believe that by systematically studying the definition, quantification, and application of heterogeneity, future MARL research will more profoundly understand the complex collaboration mechanisms between agents, and pave the way for more intelligent and adaptive collective decision-making systems.

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

## A  Limitations

Although our proposed heterogeneity distance can effectively quantify agent heterogeneity and identify various potential heterogeneities, there remain some limitations in its practical implementation. One limitation is in scaling with the number of agents. Typically, the heterogeneity distance quantification algorithm outputs a heterogeneity distance matrix for the entire multi-agent system, with a computational complexity of $O(N^2)$. When the number of agents increases significantly, matrix computation becomes costly. However, if only studying heterogeneity between specific agents in the MAS is required, the method remains effective. One only needs to remove data from other agents during CVAE training and sampling computation.

Additionally, the practical algorithms for heterogeneity quantification are built on the assumption that agent-related variables are vectors. If certain agent variables, such as observation inputs, are multimodal, operations like padding in the proposed algorithm become difficult to implement. But this does not affect the correctness of the theory. As the relevant theory still holds in this situation, additional tricks are needed for practical calculation implementation.

## B  Broader Impacts

Our work systematically analyzes heterogeneity in MARL, which has strong correlations with a series of works in MARL. Under our theoretical framework, research on agent policy diversity in MARL can be categorized within the domain of policy heterogeneity. Our work can give a new perspective for studying policy diversity. Our proposed quantification methods can not only help these works with policy evolution analysis but also explain the relationship between policy diversity and agent heterogeneity. Furthermore, our proposed HetDPS, as an application case, can also be classified among parameter sharing-based works.

Additionally, some traditional heterogeneous MARL works can be categorized within environment-related heterogeneity domains. Our quantification and definition methods are orthogonal to these works, which can fully utilize our proposed methodology for further advancement. For instance, observation heterogeneity quantification can be used to enhance agents' ability to aggregate heterogeneous observation information; transition heterogeneity quantification can help design intrinsic rewards to assist heterogeneous multi-agents in learning cooperative policies.

In conclusion, our work not only expands the scope of heterogeneity in MARL but also closely connects with many current hot topics, contributing to the further development of these works.

## C  An introduction to POMG

Partially Observable Markov Game (POMG) is essentially an extension of Partially Observable Markov Decision Process (POMDP), which in turn extends Markov Decision Process (MDP). MDP [Bellman, 1957, Kaelbling et al., 1996] is a mathematical framework that describes sequential decision-making by a single agent in a fully observable environment. In an MDP, the agent can fully observe the environment's state, select actions based on the current state, and aim to maximize cumulative rewards. Compared to MDP, the key extension of POMDP [Kaelbling et al., 1998, Cassandra, 1998] is the consideration of partial observability, making it suitable for modeling both single-agent partially observable problems [Spaan, 2012] and multi-agent problems [Bernstein et al., 2002, Oliehoek et al., 2016]. In multi-agent POMDPs, agents typically operate in a fully cooperative mode, where their rewards are usually team-shared.

The key extension of POMG over POMDP lies in modeling mixed game relationships among multiple agents. Unlike POMDP, agents in POMG do not share a common reward function; instead, each agent has its own (agent-level) reward function, making POMG more general [Sun et al., 2023, Gronauer and Diepold, 2022]. This design enables POMG to handle competitive, cooperative, and mixed interaction scenarios, better reflecting the complexity of real-world multi-agent systems. The logical relationships among Markov decision processes and their variants are illustrated in Figure 7 and Figure 8. As shown in these figures, POMG is the most general framework for modeling original problems in the MARL domain. For these reasons, we chose POMG as the foundation for discussing heterogeneity in MARL.

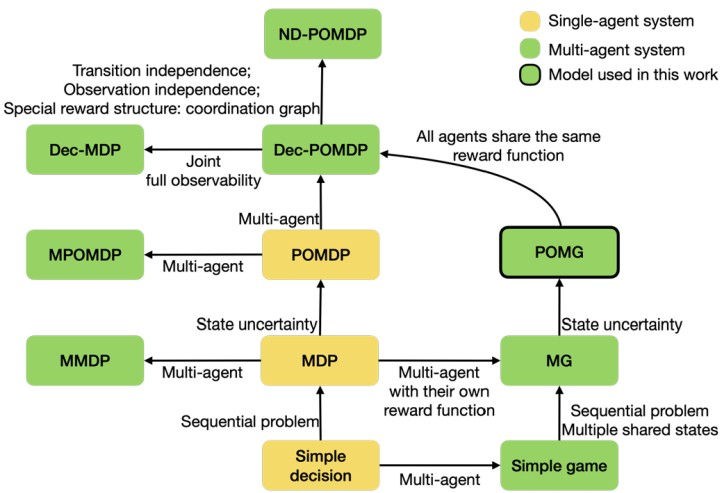

Figure 7: Common multi-agent problem formulations [Kochenderfer et al., 2022].

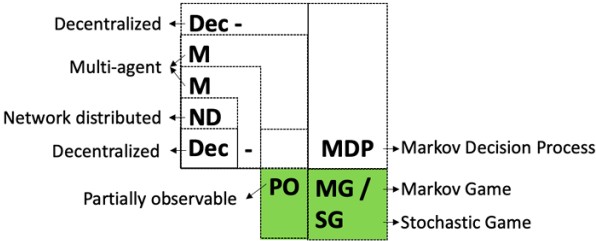

Figure 8: Common nomenclature for multi-agent models [Sun et al., 2023].

# D  Other potential types of heterogeneity in MARL

Benefiting from the reinforcement learning modeling based on POMG, we have clearly defined the boundaries of heterogeneity discussed in this paper. In fact, within the realm of unconventional multi-agent systems, there might be other types of heterogeneity.

For instance, agents may have different length of decision timesteps, with some agents inclined towards long-term high-level decisions, while others tend to make short-term low-level decisions. Agents may also have different discount factors, some works try to assign varying discount factors to different agents during algorithm training [Nguyen et al., 2022], to encourage agents to develop "*my-opic*" or "*far-sighted*" policy behaviors, thereby promoting agent cooperation. However, differences in discount factors are more reflective of algorithmic design variations rather than environmental distinctions, and thus fall outside the scope of this paper. Moreover, there may be heterogeneity among agents regarding communication, agents might have different communication channels due to hardware variations. However, the establishment of communication protocols aims to enable agents to receive more information when making decisions, potentially overcoming non-stationarity and partial observability issues [Gronauer and Diepold, 2022]. These communication messages are essentially mappings of global information processed in the environment, which are then input into the action-related network modules. From this perspective, agent communication can be modeled as a more generalized observation function that maps global information to local observations for agent decision-making, and communication heterogeneity can be categorized under observation heterogeneity. From a learning perspective, agents might also have heterogeneous available knowledge, such as differences in initial basic policies or variations in supplementary knowledge accessible during execution phase. Moreover, heterogeneity might extend beyond abstract issues, including computational resource differences among agents during learning.

Overall, even from the perspective of multi-agent reinforcement learning, heterogeneity in multi-agent systems remains a domain with extensive discussion space, warranting further subsequent research.

## E    Properties of heterogeneity distance

**Recap.** The heterogeneity distance between two agents in Section 4 can be computed as follows:

$$d_{ij}^F = \int_{x \in X} D[F_i(\cdot|x), F_j(\cdot|x)] \cdot p(x) \, dx, \tag{6}$$

where $X$ is the space of independent variables, $p(x)$ is the probability density function, and $D[\cdot, \cdot]$ is a measure that quantifies the difference between distributions.

**Proposition 1**. (*Properties of Heterogeneity Distance*) ① *Symmetry*: $d_{ij}^F = d_{ji}^F$; ② *Non-negativity*: $d_{ij}^F \geq 0$; ③ *Identity of indiscernibles*: $d_{ij}^F = 0$ if and only if agents $i$ and $j$ are $F$-homogeneous; ④ *Triangle inequality*: $d_{ij}^F \leq d_{ik}^F + d_{kj}^F$ $(i, j, k \in N)$. This proposition holds as long as the measure $D$ satisfies Property ①②③④.

**Proof.** It can be proven that when $D$ satisfies Property ①②③④, heterogeneity distance also satisfies Property ①②③④.

*1) Proof of Symmetry:*

$$d_{ij}^F = \int_{x \in X} D\left[F_i(\cdot|x), F_j(\cdot|x)\right] \cdot p(x) dx = \int_{x \in X} W\left[F_j(\cdot|x), F_j(\cdot|x)\right] \cdot p(x) dx = d_{ji}^F. \tag{7}$$

*2) Proof of Non-negativity:*

$$d_{ij}^F = \int_{x \in X} D\left[F_i(\cdot|x), F_j(\cdot|x)\right] \cdot p(x) dx \geq \int_{x \in X} 0 \cdot p(x) dx = 0. \tag{8}$$

*3) Proof of Identicals of indiscernibility (necessary conditions):*

if agent $i$ and agent $j$ are $F$-homogeneous, then we have: $X^{(i)} = X^{(j)}, \forall x \in X = X^{(i)}, F_i(\cdot|x) = F_j(\cdot|x)$,

$$
\begin{aligned}
d_{ij}^F &= \int_{x \in X} D\left[F_i(\cdot|x), F_j(\cdot|x)\right] \cdot p(x) dx \\
&= \int_{x \in X} D\left[F_i(\cdot|x), F_i(\cdot|x)\right] \cdot p(x) dx \\
&= \int_{x \in X} 0 \cdot p(x) dx \\
&= 0.
\end{aligned}
\tag{9}
$$

*4) Proof of Identicals of indiscernibility (sufficient conditions):*

$$
\begin{aligned}
d_{ij}^F = 0 &\xrightarrow{\text{Prop.②}} D\left[F_i(\cdot|x), F_i(\cdot|x)\right] = 0, \forall x \in X^{(i)} or X^{(j)} \\
&\xrightarrow{\text{Prop.② of } D} F_i(\cdot|x) = F_i(\cdot|x), \forall x \in X, X = X^{(i)} = X^{(j)},
\end{aligned}
\tag{10}
$$

then we have agent $i$ and agent $j$ are $F$-homogeneous.

*5) Proof of Triangle Inequality:*

$$
\begin{aligned}
d_{ij}^F &= \int_{x \in X} D\left[F_i(\cdot|x), F_j(\cdot|x)\right] \cdot p(x) \, dx \\
&\leq \int_{x \in X} \left(D\left[F_i(\cdot|x), F_k(\cdot|x)\right] + D\left[F_k(\cdot|x), F_j(\cdot|x)\right]\right) \cdot p(x) \, dx \\
&= \int_{x \in X} D\left[F_i(\cdot|x), F_k(\cdot|x)\right] \cdot p(x) \, dx + \int_{x \in X} D\left[F_k(\cdot|x), F_j(\cdot|x)\right] \cdot p(x) \, dx \\
&= d_{ik}^F + d_{kj}^F.
\end{aligned}
\tag{11}
$$

In this paper, we choose the *Wasserstein Distance* [Vaserstein, 1969] as the metric to quantify the distance between distributions, which satisfies the property ①②③④ [Bettini et al., 2023b].

**Discussion.** In practical computation, we adopt a representation learning-based approach to find an alternative latent variable distribution $p_i(z|x)$ to replace the original distribution $F_i(y|x)$ for quantification. It can be easily proved that when using latent variable distributions to compute heterogeneous distances, these distances still satisfy properties ①, ②, and ④ (following the same proof method as above).

In the model-based case, $p_i(z|x) = f_\phi(y_i, x)$, where $f_\phi$ represents the encoder of the CVAE. When two agents have the same independent and dependent variables (identical agent functions), their latent variable distributions are also identical. In this case, it is straightforward to prove that property ③ still holds under the model-based case.

In the model-free case, $p_i(z|x) = f_\phi(i, x)$. Due to the lack of an environment model, even agents with identical mappings may learn different representation distributions through their encoders, thus not satisfying property ③. However, as demonstrated in Section 4.2, although we cannot strictly determine agent homogeneity using $d_{ij}^F = 0$, the heterogeneity distances measured between homogeneous agents in the model-free case are sufficiently small. Moreover, the model-free manner is adequate to distinguish between homogeneous and heterogeneous agents, and still maintains the ability to quantify the degree of heterogeneity (as shown in Sections 4.2 and 6).

# F   More details of computing heterogeneity distance

Here, we present five formulas for calculating heterogeneity distances, corresponding to the five types of heterogeneity discussed in this paper.

Regarding **observation heterogeneity**, its relevant elements include the agent's observation space and observation function. For two agents $i$ and $j$, let their observation heterogeneity distance be denoted as $d_{ij}^\Omega$. The corresponding calculation formula is:

$$d_{ij}^\Omega = \int_{\hat{s} \in \{S^i\}_{i \in N}} D\left[\Omega_i(\cdot|\hat{s}), \Omega_j(\cdot|\hat{s})\right] \cdot p(\hat{s}) \, d\hat{s}, \tag{12}$$

where $D[\cdot, \cdot]$ represents a measure of distance between two distributions, and $p(\cdot)$ is the probability density function (this notation applies to subsequent equations). Here, $\hat{s}$ denotes the global state, $\{S^i\}_{i \in N}$ represents the global state space, and $\Omega_i$ and $\Omega_j$ are the observation functions of agents $i$ and $j$, respectively.

Regarding **response transition heterogeneity**, its relevant elements include the agent's action space, state space, and global state transition function. For two agents $i$ and $j$, let their response transition heterogeneity distance be denoted as $d_{ij}^\mathcal{T}$. The corresponding calculation formula is:

$$d_{ij}^\mathcal{T} = \int_{\hat{s} \in \{S^i\}_{i \in N}} \int_{\hat{a} \in \{A^i\}_{i \in N}} D\left[\mathcal{T}^i(\cdot|\hat{s}, \hat{a}), \mathcal{T}^j(\cdot|\hat{s}, \hat{a})\right] \cdot p(\hat{s}, \hat{a}) \, d\hat{a} d\hat{s}, \tag{13}$$

where $p(\cdot, \cdot)$ represents the joint probability density function. $\hat{s}$ and $\hat{a}$ denote the global state and global action respectively, $\{S^i\}_{i \in N}$ and $\{A^i\}_{i \in N}$ represent the global state space and global action space, and $\mathcal{T}_i$ and $\mathcal{T}_j$ are the local state transition functions of agents $i$ and $j$, respectively.

Regarding **effect transition heterogeneity**, its relevant elements include the agent's action space, state space, and global state transition function. For convenience, we denote $S^{-i} = \times_{k \in N, k \neq i} S^k \times S^E$ as the joint state space of all agents except agent $i$, $A^{-i} = \times_{k \in N, k \neq i} A^k$ as the joint action space of all agents except agent $i$, and $\mathcal{T}^{-i}$ as the collection of state transitions excluding agent $i$. For two agents $i$ and $j$, let their effect transition heterogeneity distance be denoted as $d_{ij}^{\mathcal{T}^-}$. The corresponding calculation formula is:

$$d_{ij}^{\mathcal{T}^-} = \int_{s' \in S^{(-i)}} \int_{s \in A^i} \int_{a' \in A^{(-i)}} \int_{a \in A^i} D\left[\mathcal{T}^{-i}(\cdot|x), \mathcal{T}^{-j}(\cdot|x)\right] \cdot p(x) da da' ds ds', \tag{14}$$

where for convenience, we denote $x = (s', s, a', a)$, and $p$ is the joint probability density function.

The calculation of effect transition heterogeneity distance differs from the previous two types of heterogeneity distances in two significant ways. The first difference lies in its introduction of agent-

level elements as variables rather than global variables. When two agents have different agent-level variable spaces, it becomes challenging to calculate the heterogeneity distance under this definition. The second difference is that it involves a quadruple integral, making its computational complexity much higher than the single or double integrals of the previous two distances.

These two differences make the calculation of effect transition heterogeneity distance more challenging. Fortunately, through our proposed meta-transition model, we can simplify the calculation of effect transition heterogeneity distance to a double integral that only involves the agent's local states and actions. Additionally, the distance measurement through representation learning also reduces the constraints on the similarity of agents' variable spaces. Even when two agents have different variable spaces (for example, one agent's local state space is 10-dimensional while another's is 20-dimensional), we can still process the variable inputs through techniques like padding and then map them to the same dimension using encoder networks. This demonstrates that the approach based on representation learning and meta-transition significantly extends the applicability of heterogeneity distance measurement, which also holds true in the quantification of heterogeneous types discussed below.

Regarding **objective heterogeneity**, its relevant element is the agent's reward function. For two agents $i$ and $j$, let their objective heterogeneity distance be denoted as $d_{ij}^r$. The corresponding calculation formula is:

$$d_{ij}^r = \int_{\hat{s} \in \{S^i\}_{i \in N}} \int_{\hat{a} \in \{A^i\}_{i \in N}} D\left[r^i(\cdot|\hat{s}, \hat{a}), r^j(\cdot|\hat{s}, \hat{a})\right] \cdot p(\hat{s}, \hat{a}) \, d\hat{a} d\hat{s}, \tag{15}$$

where $p(\cdot, \cdot)$ represents the joint probability density function. $\hat{s}$ and $\hat{a}$ denote the global state and global action respectively, $\{S^i\}_{i \in N}$ and $\{A^i\}_{i \in N}$ represent the global state space and global action space, and $r_i$ and $r_j$ are the reward functions of agents $i$ and $j$, respectively.

Regarding **policy heterogeneity distance**, its relevant elements include the agent's observation space, action space, and policy function. For two agents $i$ and $j$, let their policy heterogeneous distance be denoted as $d_{ij}^\pi$. The corresponding calculation formula is:

$$d_{ij}^\pi = \int_{o \in O^i} D\left[\pi_i(\cdot|o), \pi_j(\cdot|o)\right] \cdot p(o) \, do, \tag{16}$$

where $D[\cdot, \cdot]$ represents a measure of distance between two distributions, and $p(\cdot)$ is the probability density function. Here, $o$ denotes the observation, $O^i$ represents the observation space, and $\pi_i$ and $\pi_j$ are the policy functions of agents $i$ and $j$, respectively.

# G    Meta-Transition and its Heterogeneity Distance

To quantify an agent's comprehensive heterogeneity, we introduce the concept of meta-transition. Meta-transition is a modeling approach that explores an agent's own attributes from its perspective. Our goal is to quantify an agent's comprehensive heterogeneity using only the agent's local information (as global information is typically difficult to obtain in practical MARL scenarios).

Based on this, we provide the definition of meta-transition. Let the meta-transition of agent $i$ be denoted as $M_i$. It is a mapping $M_i : S_i \times A_i \rightarrow S_i \times R \times \Omega_i$. At time step $t$, the inputs of meta-transition are the agent's local state $s_t^i$ and local action $a_t^i$, and the outputs are the next time step's local state $s_{t+1}^i$, the next time step's local observation $o_{t+1}^i$, and the current time step's reward $r_t^i$ based on the state and action.

We explain why the above relationship can reflect all agent-level elements in POMG. The input local state and local action of meta-transition actually correspond to the inverse mapping to the global state and global action. This inverse mapping potentially restores the local state and action to global information, and then obtains the next time step's global state according to the global state transition function, which is mapped to local observation through the observation function. Therefore, this process reflects the agent's effect transition heterogeneity and observation heterogeneity. Additionally, the potential global state and global action also determine the agent's local state and corresponding reward at the next time step, which reflect the agent's response transition heterogeneity and objective heterogeneity, respectively.

It is worth noting that meta-transition is not a function that actually exists in POMG, but an implicitly defined mapping. We aim to quantify this mapping difference to capture the agent's comprehensive heterogeneity. Therefore, meta-transition heterogeneity is quantified in a model-free manner.

Moreover, meta-transition is not limited to the aforementioned form. It can be transformed into different forms according to the modular settings of independent and dependent variables. For example, by removing the agent's reward, meta-transition can reflect the agent's observation heterogeneity, response transition heterogeneity, and effect transition heterogeneity.

After determining the input and output of meta-transition, the relevant heterogeneity distance can be calculated using the same model-free method as before. Since meta-transition involves multiple variables, and the dimensions between these variables may differ significantly (for example, the dimension of reward is 1, while the dimension of observation might be 100), directly fitting with deep networks may struggle to capture information corresponding to low-dimensional variables. We address this issue through a dimension replication trick. In practice, we typically replicate the reward dimension to be similar to the dimensions of observation or action, ensuring that the autoencoder network can capture information related to objective heterogeneity during learning.

## H   Derivation of ELBO

The Evidence Lower Bound (ELBO) of the likelihood can be derived as follows:

$$
\begin{aligned}
\log p(y|x) &= \log \int p(y, z|x) dz &\text{(a)}\\
&= \log \int \frac{p(y, z|x) f_\phi(z|y, x)}{f_\phi(z|y, x)} dz &\text{(b)}\\
&= \log \mathbb{E}_{f_\phi(z|y,x)} \left[ \frac{p(y, z|x)}{f_\phi(z|y, x)} \right] &\text{(c)}\\
&\geq \mathbb{E}_{f_\phi(z|y,x)} \left[ \log \frac{p(y, z|x)}{f_\phi(z|y, x)} \right] &\text{(d)}\\
&= ELBO_{\text{model-based}},
\end{aligned}
\tag{17}
$$

where $f_\phi(z|y, x)$ represents the posterior probability distribution of the latent variable generated by the encoder, and $p(y, z|x)$ denotes a joint probability distribution concerning the customized feature and latent variable, conditioned on $o$. Throughout the derivation of the formula, (a) employs the properties of the joint probability distribution, (b) multiplies both numerator and denominator by $f_\phi(z|y, x)$, (c) applies the definition of mathematical expectation, and (d) invokes the Jensen's inequality.

Considering that the ELBO includes an unknown joint probability distribution, we can further decompose it by using the posterior probability distributions from the encoder and decoder:

$$
\begin{aligned}
ELBO_{\text{model-based}} &= \mathbb{E}_{f_\phi(z|y,x)} \left[ \log \frac{p(y, z|x)}{f_\phi(z|y, x)} \right]\\
&= \mathbb{E}_{f_\phi(z|y,x)} \left[ \log \frac{g_\omega(c|z, x) p(z|x)}{f_\phi(z|y, x)} \right] &\text{(a)}\\
&= \mathbb{E}_{f_\phi(z|y,x)} \left[ \log g_\omega(c|z, x) \right]\\
&\quad + \mathbb{E}_{f_\phi(z|y,x)} \left[ \log \frac{p(z|x)}{f_\phi(z|y, x)} \right] &\text{(b)}\\
&= \underbrace{\mathbb{E}_{f_\phi(z|y,x)} \left[ \log g_\omega(c|z, x) \right]}_{\text{reconstruction term}} - \underbrace{D_{\text{KL}} \left[ f_\phi(z|y, x) \| p(z|x) \right]}_{\text{prior matching term}}, &\text{(c)}
\end{aligned}
\tag{18}
$$

where $f_\phi(z|y, x)$ and $g_\omega(c|z, x)$ are the posteriors from the encoder and decoder, respectively. The conditional joint probability distribution $p(y, z|x)$ is a imaginary construct in mathematical terms and lacks practical significance. It can be formulated using the probability chain rule, constructed from the

posterior distribution of the customized feature and the prior distribution of the latent variable (step (a)). Step (b) decomposes the expectation, and step (c) applies the definition of the KL divergence.

Thus, the ELBO can be decomposed into a reconstruction term of the customized feature, and a prior matching term of the posterior and the prior. By maximizing the ELBO, the reconstruction likelihood can be maximized while minimizing the KL divergence between the posterior and the prior. In the model-free case, the same approach can be used to derive the ELBO and corresponding loss function.

# I  Details of HetDPS

HetDPS is a novel algorithm designed to efficiently manage the allocation of neural network parameters across multiple agents in MARL. This algorithm leverages the Wasserstein distance matrix to cluster agents based on their similarities, and subsequently assigns them to suitable neural networks. The pseudocode of HetDPS is shown in Algorithm 1.

The algorithm begins by computing the affinity matrix from the Wasserstein distance matrix, which is then used as input to the Affinity Propagation clustering algorithm. This process yields a new set of cluster assignments for the agents. If it is the first time the algorithm is executed, the cluster assignments are directly used as network assignments.

In subsequent iterations, the algorithm compares the new cluster assignments with the previous ones to determine the optimal network assignments. This is achieved by constructing an overlap matrix that captures the similarity between the old and new cluster assignments. Based on the number of old and new clusters, the algorithm handles three distinct cases:

1. Equal number of old and new clusters: In this scenario, the algorithm establishes a one-to-one mapping between the old and new clusters using the Hungarian algorithm. It then constructs a mapping from old clusters to networks and assigns each agent to a network based on its new cluster assignment.

2. More new clusters than old clusters: When the number of new clusters exceeds the number of old clusters, the algorithm handles network splitting. It uses the Hungarian algorithm to find the best matching between old and new clusters and establishes a mapping from new clusters to old clusters. For new clusters without a clear match, the algorithm either finds the most similar old cluster or identifies the closest network. It then executes a splitting operation to copy parameters from the source network to the new network.

3. More old clusters than new clusters: In this case, the algorithm handles network merging. It uses the Hungarian algorithm to find the best matching between old and new clusters and establishes a mapping from old clusters to new clusters. For each new cluster, it identifies the networks to be merged and executes a merging operation based on the specified merge mode (majority, random, average, or weighted). The algorithm then assigns each agent to a network based on its new cluster assignment.

HetDPS offers a flexible and efficient approach to managing neural network parameters in multi-agent systems. By dynamically adjusting network assignments based on agent similarities, the algorithm enables effective parameter sharing and reduces the need for redundant computations.

**Algorithm 1** HetDPS

```
 1: Initialize policies and parameter sharing paradigm
 2: for episode = 1 to maxEpisodes do
 3:     Interact with environment to collect data
 4:     Add data to reinforcement learning (RL) sample pool
 5:     Add data to heterogeneity distance sample pool
 6:     if episode % trainingPeriod = 0 then
 7:         Update policies using RL sample pool
 8:     end if
 9:     if episode % quantizationPeriod = 0 then
10:         Compute heterogeneity distance matrix D (Section 4)
11:         Cluster agents using Affinity Propagation on D
12:         if no previous clustering exists then
13:             Assign networks to agents based on clusters
14:             Copy network parameters as needed
15:         else
16:             Compute maximum overlap matching between current and previous clusters
17:             if number of clusters unchanged then
18:                 Map new clusters to previous networks
19:             else if new clusters > previous clusters then
20:                 Split networks: copy parameters for unmatched clusters
21:             else
22:                 Merge networks: combine parameters based on merge mode
23:             end if
24:             Assign networks to agents
25:         end if
26:     end if
27: end for
```

