# OpenReview forum: "Heterogeneity in Multi-Agent Reinforcement Learning"
_NeurIPS.cc/2025/Conference — Submitted to NeurIPS 2025_

### Official Review · Reviewer_dH7A · 2025-06-30

**Clarity:** 3
**Significance:** 4
**Originality:** 4
**Rating:** 5
**Confidence:** 4

**Summary:**

This paper presents a systematic framework for understanding and leveraging heterogeneity in MARL. The authors first categorize heterogeneity into five types—observation, response transition, effect transition, objective, and policy—based on a formal agent-level modeling under the Partially Observable Markov Game framework. Each type is given a mathematical definition. To quantify heterogeneity, the paper introduces the notion of heterogeneity distance, which measures functional differences between agents using representation learning techniques such as Conditional Variational Autoencoders, applicable in both model-based and model-free settings. Additionally, a meta-transition abstraction is also proposed to capture overall agent-level heterogeneity. Building on these foundations, the authors develop HetDPS, a dynamic parameter sharing algorithm that clusters agents based on heterogeneity distances and adjusts sharing schemes during training. This improves policy diversity while maintaining sample efficiency.

**Questions:**

1. The proposed method for computing heterogeneity distance incurs a quadratic computational cost with respect to the number of agents, which could limit its applicability in large-scale MARL scenarios. While the paper mentions partial remedies such as parallel computation and agent subset selection, these aspects are not discussed in depth in the main text.

It would be helpful if the authors could provide a more detailed empirical or theoretical analysis of the computational overhead associated with their method, especially under varying agent counts. Exploring approximate or hierarchical alternatives (e.g., sparse distance updates, local sampling) would enhance the method's practicality.

2. The proposed heterogeneity quantification relies heavily on representation learning via CVAE, but the paper does not provide an analysis of the training stability, sensitivity to hyperparameters, or generalization across tasks.

I think authors need to include an evaluation of CVAE stability, such as training convergence plots or reconstruction quality metrics. In addition, it would be valuable to assess whether learned heterogeneity representations remain meaningful under task shifts or in transfer learning settings.

3. While the paper references related works (e.g., MADPS, AdaPS), the delineation of methodological contributions, especially in relation to prior studies on policy diversity and heterogeneous grouping, could be more explicit. In detail, the paper references several existing methods, such as MADPS (which performs dynamic grouping based on policy distance) and AdaPS (which relies on response transition heterogeneity), but it does not clearly articulate in which aspects the proposed approach is superior or fundamentally different. In particular, the area of policy diversity has been extensively explored, with many prior works addressing behavioral differences among agents. Therefore, the authors need to clarify what is novel about their formulation of heterogeneity distance compared to existing studies in this domain.

I think it is necessary to establish a more structured comparison—e.g., a table summarizing key distinctions in terms of heterogeneity types addressed, theoretical definitions, and application scopes—would clarify the novelty of the proposed approach. Furthermore, incorporating controlled ablation experiments where heterogeneity quantification is removed or substituted could help isolate its contribution.

4. The current empirical validation focuses primarily on particle-based spreading and selected SMAC tasks. It remains unclear how well the proposed method generalizes across different MARL domains or tasks with diverse forms of heterogeneity.

I think it would be better to evaluate the method on additional domains (e.g., mixed cooperative-competitive environments or hierarchical tasks) would help demonstrate its broader applicability. Alternatively, providing a discussion of the method’s limitations in such settings would also be valuable.

**Ethical Concerns:**

["NO or VERY MINOR ethics concerns only"]

**Final Justification:**

I recommend accept. The paper provides a rigorous framework for heterogeneity in MARL, defining five types with formal clarity and introducing a quantification method that is both sound and broadly applicable.

The authors improved clarity by condensing definitions, moving methodological details into the main text, and providing more explicit comparisons to related work. While concerns about scalability, similarity to prior work, and limited empirical scope remain, the rebuttal provided convincing clarifications and additional experiments showing efficiency and robustness.

Although the performance gains are not dramatic, the conceptual novelty, rigorous framework, and practical insights outweigh these weaknesses. I think this work will help structure future MARL research and serves as a strong foundation for follow-up studies.

**Limitations:**

Yes.

**Paper Formatting Concerns:**

No.

**Quality:**

3

**Strengths And Weaknesses:**

The paper presents a well-structured and technically sound framework for defining, quantifying, and utilizing heterogeneity in MARL. The theoretical modeling based on the POMG framework is rigorous and covers multiple aspects of heterogeneity. The introduction of heterogeneity distance and its implementation via CVAE-based representation learning is technically well-founded. The experimental evaluation is comprehensive, including both synthetic and benchmark scenarios. The application to dynamic parameter sharing (HetDPS) further demonstrates practical utility. However, the scalability of the heterogeneity distance computation (quadratic in agent number) may limit its applicability to large-scale systems. Moreover, some methodological choices (e.g., reliance on latent space alignment) could benefit from deeper comparative baselines or ablation studies.

The paper is generally clear and well-organized. Definitions are formally presented, the taxonomy is intuitively motivated, and the figures (e.g., heterogeneity matrices and case studies) are informative. That said, certain sections (e.g., Section 4’s explanation of representation learning-based measurement) may be dense for readers not familiar with CVAE or distribution distance metrics. A clearer separation between theoretical definitions and their practical computation would improve readability.

The work addresses a foundational gap in the MARL literature: the lack of a rigorous, unified treatment of agent heterogeneity. The proposed framework has potential impact in both theoretical studies (e.g., policy analysis, agent grouping) and practical applications (e.g., efficient training via adaptive parameter sharing). The significance is further enhanced by its compatibility with both model-based and model-free settings. While the primary application (HetDPS) is within parameter sharing, broader implications for general-purpose heterogeneous MARL algorithms could be explored more explicitly.

The taxonomy of heterogeneity types and the formal notion of heterogeneity distance represent a novel contribution. Prior work has either focused on specific types (e.g., policy diversity) or has lacked rigorous definitions. The combination of formal theory with representation learning for quantification is both original and insightful. The meta-transition concept is also novel and provides a useful abstraction. While the application to parameter sharing is incremental, the underlying methodological contributions are notably innovative.

---

> ### Author Rebuttal · Authors · 2025-07-31
>
> We sincerely appreciate the detailed summary and constructive comments on our work.
>
> We are grateful that Reviewer dH7A recognized *our methodology and framework for addressing heterogeneity as being well-structured and technically sound. The reviewer's acknowledgment that our paper is generally clear and well-organized, while also commending our work for addressing a foundational gap in MARL with potential impacts across both theoretical studies and practical applications, is particularly encouraging*. Specifically, we value the reviewer's assessment that our classification and definitions of heterogeneity represent a significant contribution, that our quantification method demonstrates both originality and insightfulness, and that the meta-transition concept introduces a novel and useful abstraction.
>
> In the following sections, we will address the potential weaknesses raised by the reviewer and provide comprehensive responses to all questions posed.
>
>
>
> ---
>
>
> **R4-1** *Scalability and Ablation Studies*
>
> We appreciate your suggestion. Additional experiments have been conducted to evaluate the computational cost and performance of our method under varying scales. An ablation study was performed by removing the CVAE module to compare its performance with direct Wasserstein Distance computation.
>
> Results demonstrate that while our method does incur higher computational costs at scale (as acknowledged in the Limitations section), it exhibits superior scalability compared to alternative approaches while achieving better quantification performance. Experimental details are provided in R3-4. The computational curve exhibits a peak-like pattern, contrasting with the stable state observed in our approach.
>
> **R4-2** *Enhanced Readability*
>
> We acknowledge the reviewer's concern regarding the density of certain sections for readers unfamiliar with the background. Given the extensive content and page constraints, we have compiled all relevant background knowledge in the appendix and supplementary materials. Per your suggestion, we have expanded these explanatory sections in the appendix.
>
> **R4-3** *General-Purpose Heterogeneous MARL Algorithms*
>
> Indeed! Our method demonstrates compatibility with diverse MARL applications. Notably, we have explicitly discussed in the Broader Impact section how it interfaces with other domains—such as heterogeneous observation aggregation and intrinsic reward design. We sincerely appreciate this suggestion and plan to further explore these directions as future work.
>
>
> ---
>
> **Q4-1 A**:  Thanks. We have already answered your question in R4-1.
>
> **Q4-2 A**: Thank you very much for your suggestion. However, due to time constraints, we cannot conduct further research on cross-task transfer. Our research focuses on the heterogeneity of MARL, and from this perspective, it is already a comprehensive work. In the early stages of our research, we conducted thorough convergence experiments on CVAE, and the experiments showed that the Loss had significantly decreased by the 2nd epoch, and the Loss curve had converged by the 5th epoch. To ensure sufficient training, we set the number of epochs to 10; to prevent overfitting in a specific quantization cycle, we chose to retain the CVAE model and inherit it to the next quantization cycle for training with new samples.
>
> **Q4-3 A**: Thank you very much for your suggestion. In fact, we have fully discussed articles on heterogeneity, parameter sharing, and policy diversity in MARL in the Related Work section and defined their scope. Details about this part can be found in the supplementary material.
>
>
> **Q4-4 A:** This is an interesting question. Although our method focuses on the heterogeneity of agents, the related quantification methods can be applied to other scenarios. For example, they can be used to quantify the differences between MDPs or the differences between opponent agents. We need to point out that existing methods explicitly only target quantifying the heterogeneous distance within a single collaborative multi-agent system and, are therefore only applicable to MARL tasks that can be modeled as fully collaborative.
>
> Thank you for your inspiration, we will consider this as a future research direction.
>
> ---
>
>
> Finally, we sincerely appreciate the reviewer's time and effort during the review process. Finally, we hope that the above responses address your concerns. If you have any thoughts or suggestions, we welcome further communication! We look forward to your reply.

---

> > ### Comment · Reviewer_dH7A · 2025-08-07
> >
> > Thank you for the detailed and thoughtful rebuttal. I appreciate the authors' clarifications regarding scalability experiments, the stability of CVAE module, and the theoretical research on heterogeneous agents. The empirical results are strong, and the proposed framework shows promise in leveraging heterogeneity for efficient MARL exploration. That said, I still believe there remain limitations that merit further investigation—transfer learning,more experiments on other cooperative benchmarks. I also agree with some concerns raised by other reviewers regarding the generality of the approach and the discussion of related work. Therefore, I acknowledge the authors' efforts and the strengths of the work and maintain my original rating of accept. I believe the paper contributes valuable ideas and empirical findings, even though certain aspects could benefit from additional refinement.

---

> > > ### Author Response · Authors · 2025-08-08
> > >
> > > We appreciate your response. We are pleased that the reviewer maintained a positive initial score. We are also grateful that the reviewers acknowledged the strength of our empirical results and the promising impact of our proposed framework on the  MARL domain. Your suggestions are very helpful for us to carry out future work.
> > >
> > > Additionally, we appreciate the detailed suggestions from the other reviewers. During the rebuttal and discussion phase, we have followed the reviewers' suggestions to simplify the definition section, move details of methodology from the appendix to the main text, and provide a more explicit comparison with related works. Although the refinements are relatively minor, they have significantly improved the clarity of our paper.
> > >
> > >
> > > Finally, we once again sincerely thank you for your effort and constructive discussion.

---

> ### Comment · Area_Chair_DVhv · 2025-08-05
>
> Thanks to the authors for their rebuttal! Since this paper got borderline reviews, a critical evaluation whether the rebuttal addresses the reviewer's questions and criticisms is very important. Could the reviewer please check whether the rebuttal still left them with open questions, or just acknowledge that this is not the case?

---

### Official Review · Reviewer_21mt · 2025-07-01

**Clarity:** 2
**Significance:** 2
**Originality:** 2
**Rating:** 3
**Confidence:** 4

**Summary:**

This paper presents a comprehensive study of heterogeneity in multi-agent reinforcement learning, including its definition, quantification, and utilization. The authors first systematically categorizes heterogeneity into five types: observation, response transition, effect transition, objective, and policy heterogeneity. Further, it introduces a heterogeneity distance and a representation learning-based method to quantify heterogeneity in both model-free and model-based settings, along with the concept of meta-transition heterogeneity to capture comprehensive agent differences. Building on this quantification, a multi-agent dynamic parameter-sharing algorithm is proposed, offering improved interpretability and reduced reliance on task-specific hyperparameters compared to existing methods.

**Questions:**

1. What are the differences between the proposed heterogeneity distance in the paper and previous methods?

2. Can the heterogeneity between agents lead to performance improvement?

3. Can other distance metrics affect the performance of the proposed method?

4. Why are some state-of-the-art MARL methods, such as QMIX (with full parameter sharing), not compared in the experimental section?

**Ethical Concerns:**

["NO or VERY MINOR ethics concerns only"]

**Final Justification:**

This paper presents a complete definition of heterogeneity in MARL. However, the quantification of heterogeneity has been thoroughly discussed in previous works, as listed in the review, which decreases the novelty of the work.

**Limitations:**

yes

**Paper Formatting Concerns:**

The font size in the figures presented in the experimental section is rather small.

**Quality:**

2

**Strengths And Weaknesses:**

Strengths:
1. The authors conduct an in-depth study of heterogeneity in multi-agent reinforcement learning. They novelly categorize heterogeneity into five types: observation, response transition, effect transition, objective, and policy heterogeneity. This is a departure from the pure policy heterogeneity definition in previous works. These five types comprehensively summarize the dimensions of heterogeneity in multi-agent reinforcement learning.

2. The authors propose a standard objective to measure heterogeneity distance and introduce two practical solutions to compute it efficiently. First, standardizing the original distributions maps the intractable distributions to measurable distributions. Second, sampling based on agent-environment interactions builds a trajectory sample pool to reduce computational cost.

3. Based on the heterogeneity distance, the authors further propose a Heterogeneity-based Multi-Agent Dynamic Parameter Sharing algorithm, which groups agents into different clusters and periodically quantifies heterogeneity.

Weaknesses:

1. While the authors introduce a heterogeneity distance objective, there are many possible distance metrics that could be used. Comparing multiple distance metrics could improve the generality and robustness of the proposed method.

2. The concept of heterogeneity distance is not entirely novel and has been discussed in prior works [1,2]. However, the authors do not cite these in the paper. The techniques of standardizing original distributions and dynamic parameter sharing bear strong similarity to [1], and the differences should be clearly discussed.

3. The proposed method does not appear to demonstrate significant performance improvement over baseline methods on the SMAC tasks.

4. The operation of standardizing original distributions may be computationally expensive. In general, Wasserstein distance can be computed more efficiently using Fenchel-Rockafellar duality with trajectory sampling.

[1] Hu, T., Pu, Z., Ai, X., Qiu, T., & Yi, J. (2024). Measuring Policy Distance for Multi-Agent Reinforcement Learning. arXiv preprint arXiv:2401.11257.

[2] Bettini, M., Kortvelesy, R., & Prorok, A. (2024, July). Controlling Behavioral Diversity in Multi-Agent Reinforcement Learning. In International Conference on Machine Learning (pp. 3611-3636). PMLR.

---

> ### Author Rebuttal · Authors · 2025-07-31
>
> Thanks for your detailed comments!
>
> We appreciate that Reviewer 21mt recognizes *our research on heterogeneity as in-depth*, *our classification of heterogeneity as breaking through the limitations of previous work*, *our proposed heterogeneous distance and practical solution as efficient*, and *praises our methodology of quantifying heterogeneity through HetDPS*.
>
> Most of the concerns raised by the reviewer stem from misunderstandings about our work. We will clarify these points one by one below and answer related questions, hoping to achieve a constructive discussion with the reviewer.
>
> ---
>
> **R3-1** *Comparison of Various Distance Metrics*
>
> We assume that the distance metric mentioned by the reviewer refers to the measure *D* in equation-3 used internally in the method, rather than the heterogeneous distance method itself. According to our proposition-1 and the subsequent design of the HetDPS method, the measure *D* itself needs to satisfy distance-like properties.
>
> Therefore, we excluded divergence-based measures such as KL-divergence, which do not satisfy symmetry and the triangle inequality. To explore the impact of different measures on our method, we selected three distance-style measures and compared the performance of the HetDPS algorithm in particle scenarios.
>
> The distances we chose are: BD (Bhattacharyya Distance), HD (Hellinger Distance), and WD (Wasserstein Distance).
>
> We use normalized values to represent the scores of each metric, assuming that the WD-based curve scores 100 points and the worst Baseline curve scores 0 points, obtained through a linear function.
>
> | Distance Metric | 15a_3c | 30a_3c | 15a_5c | 30a_5c |
> |-|-|-|-|-|
> | BD | 94.37  | 98.52  | 101.89 | 96.14  |
> | HD | 100.00 | 100.00 | 100.00 | 100.00 |
> | WD    | 95.85  | 100.12 | 97.59  | 102.94 |
>
>
> Looking at both the normalized scores and the actual grouping results, the choice of measure does not affect our method's ability to achieve good performance and correct grouping.
>
> Thanks to the reviewer's suggestion, **we have added this content to the paper.**
>
> **R3-2** *References and Discussions with Other Methods*
>
> **First, we have already cited the prior works [1,2] mentioned by the reviewer in both the introduction chapter and the Related Work section (in the supplementary materials).** We have also cited work [3], which is the predecessor of the specific strategy diversity quantification method in work [2].
>
> **Second, we clarify that our work differs fundamentally from the above works.**
>
> Compared to the quantification method in [1], our method has the following differences:
>
> 1. When computing heterogeneous distances, our method fully considers the probability density of each independent variable, while method [1] does not consider this probability density;
>
> 2. Our method uses samples from MARL training as the independent variable pool, while method [1] uses samples from the entire space as the independent variable pool;
>
> 3. Our method is applicable in model-free scenarios and can measure heterogeneity beyond policy heterogeneity; method [1] can only be applied to measure policy distances in scenarios with explicit models and homogeneous inputs and outputs.
>
> Compared to the parameter sharing method in [1], our method has the following differences:
>
> 1. Our method can use various heterogeneous distances for dynamic grouping, while method [1] can only use policy distances to change policies, easily falling into the trap of circular dependency;
>
> 2. Our method fully considers the "smooth" transition of multi-agent policies, using clustering algorithms and bipartite graph matching algorithms, reducing dependence on hyperparameters and being compatible with various parameter sharing initialization forms; method [1] does not use allocation algorithms, has high requirements for policy parameter sharing initialization, and relies on task-specific hyperparameters.
>
> Compared to the quantification methods in [2,3], our method has the following differences:
>
> 1. When computing heterogeneous distances, our method fully considers the probability density of each independent variable, while methods [2,3] do not consider this probability density;
>
> 2. Our method makes no assumptions about the distribution of the model, while methods [2,3] can only quantify policy distances with assumed distributions (such as assuming actions follow a Gaussian distribution);
>
> 3. Our method is also applicable in model-free scenarios and can measure heterogeneity beyond policy heterogeneity; methods [2,3] can only be applied to measure in scenarios with explicit models.
>
>
> **R3-3** *HetDPS Shows No Significant Performance Improvement in SMAC Tasks*
>
> In fact, we do not expect our algorithm to show significant performance improvements compared to the baseline. After all, these algorithms are merely parameter sharing methods rather than complete MARL algorithms themselves. They share a common upper limit, which is to find the appropriate parameter sharing paradigm in the scenario.
>
> **The advantage of our algorithm lies in reaching this upper limit faster and more stably.** Moreover, as stated in our paper, the greatest advantage of our algorithm compared to others is its **strong interpretability and robustness to hyperparameters**. Details about this can be found in Section-B of the supplementary materials.
>
>
> **R3-4** *Concerns About Computational Expense of the Proposed Method*
>
> Introducing standardizing original distributions does indeed increase computational cost. However, after completing CAVE training, it simplifies the original complex distance-metric calculation into distance operations of Gaussian distributions that can be directly analytically calculated (and we have optimized this step). This step greatly improves computational speed.
>
> First, we have already compared the training speed of our method with all parameter sharing baseline methods in the experimental chapter. (To ensure fairness, they all performed calculations using the same amount of data over the same period), and the conclusion is that our method does not consume more computational time compared to other methods, but rather saves time compared to some algorithms.
>
> Second, we compared the time cost of our method with directly using distance-metrics without learning CVAE. We both used WD-based quantification, and compared two methods: one is 1-WD (discrete approximation), and the other is the Fenchel-Rockafellar duality method (FR) mentioned by the reviewer.
>
> We conducted experiments with different numbers of agents (computation time (second)), and the results are as follows:
>
> Comparison of time consumption for different distance calculation methods (s)
>
> | Distance Method | 4 | 10 | 20 | 40 | 100 |
> |-|-|-|-|-|-|
> | Ours | 17.13 | 34.41 | 100.01 | 190.87 | 4797.90 |
> | 1-WD | 2.92 | 7.00 | 45.79 | 115.46 | 4453.90 |
> | FR | 4.92 | 14.01 | 74.82 | 145.32  | 4510.00 |
>
>
> Furthermore, we conducted quantitative experiments to evaluate these methods in the 4a_2c scenario. The results demonstrate the average heterogeneity distances (meta-het) quantified by the three approaches. Our measurements include both intra-group and inter-group heterogeneity distances.
>
> | Metric | **Ours** | 1-WD| FR  |
> |--|-|-|-|
> | Intra-group heterogeneity distance | **0.23** | 0.67 | 0.34|
> | Inter-group heterogeneity distance | **2.56** | 0.56 | 0.89|
>
>
> The experimental data clearly indicates that while our method requires more computation time compared to alternatives, this performance gap diminishes as the number of agents increases. Crucially, our approach effectively quantifies agent heterogeneity distances, enabling clear agent differentiation - a capability absent in other methods.
>
> ---
>
> **Q 3-1 A**: We have already answered this question in R3-1 and R3-3. Overall, compared to previous work, our proposed method for quantifying heterogeneous distances is highly integrated with our methodology, can accommodate both model-based and model-free scenarios, and can quantify both local specific types of heterogeneity and overall heterogeneity.
>
> **Q 3-2 A**: This is a very profound question. Heterogeneity is not just a static physical property, but a dynamic property that interacts with the environment. Other research on MAS has also pointed out that heterogeneity is a fundamental property of MAS, but it may have negative impacts on system performance. If utilized properly, it can bring performance improvements. For details, please refer to reference [4].
>
>
>
> **Q 3-3 A**: As shown in our response to R3-1, as long as the distance metrics satisfy the properties mentioned in proposition-1, they do not significantly affect the performance of the method. However, it should be noted that some measures may experience numerical instability [Reference 3], so we still recommend using WD.
>
> **Q 3-4 A**: In the experiments regarding HetDPS, the baselines we compared are various parameter sharing methods. These methods can be combined with various SOTA MARL algorithms. Therefore, to ensure fairness, we combined all the compared methods with the same algorithm, namely MAPPO. Thus, NPS and FPS are the shared version and independent version of the classic MAPPO, respectively. Details and discussions on this part can be found in Section-B of our supplementary materials.
>
> ---
>
> Finally, we sincerely appreciate the reviewer's time and effort during the review process. We hope that the above responses address your concerns. If you have any thoughts or suggestions, we welcome further communication! We look forward to your reply.
>
> [1] Hu, T, et al. Measuring Policy Distance for Multi-Agent Reinforcement Learning. AAMAS-2024.
>
> [2] Bettini, et al. Controlling Behavioral Diversity in Multi-Agent Reinforcement Learning. ICML-2024
>
> [3] Bettini, et al. System neural diversity: Measuring behavioral
> heterogeneity in multi-agent learning. arXiv preprint, 2023b.
>
>
> [4] Chris Bennett. Heterogeneity in multi-agent systems. PhD thesis, University of Bristol, 2024.

---

> > ### Comment · Reviewer_21mt · 2025-08-03
> >
> > The authors have addressed most of my concerns. I will keep my score.

---

> > > ### Author Response · Authors · 2025-08-03
> > >
> > > Dear Reviewer 21mt,
> > >
> > > Thank you for your response. We are glad that the previous reply has addressed most of your concerns  at an early stage of discussion. Could you kindly provide an explicit list of any remaining unresolved concerns? This will help us provide a more detailed response and avoid potential misunderstandings.
> > >
> > > Sincerely, The Authors

---

> > > ### Author Response · Authors · 2025-08-06
> > >
> > > Dear Reviewer 21mt,
> > >
> > > I hope this message finds you well.
> > >
> > > As we approach the end of the discussion phase, we are eager to understand any specific issues that may remain unresolved. If there are any concerns that have not been fully addressed, please let us know. We are very willing to resolve any potential issues to improve our work.
> > >
> > > We look forward to your reply.

---

> > > > ### Comment · Reviewer_21mt · 2025-08-07
> > > >
> > > > We appreciate the authors' efforts. However, their discussion of the differences between this work and the method in [1] contains some inaccuracies. The method in [1] was explicitly proposed to measure policy differences between heterogeneous agents with different action spaces or policy distributions (as discussed in the third-to-last paragraph of Section 3.1 of [1]). This contradicts the authors' claim that their " method [1] can only be applied to measure policy distances in scenarios with explicit models and homogeneous inputs and outputs." Furthermore, the network structure and learning loss are quite similar to those in [1]. We encourage the authors to provide a more accurate and balanced comparison between their work and reference [1], which will further strength this work's contributions.
> > > >
> > > > [1] Hu, T, et al. Measuring Policy Distance for Multi-Agent Reinforcement Learning. AAMAS-2024.

---

> ### Author Response · Authors · 2025-08-08
>
> Thanks for your response. We are pleased that our previous responses have indeed addressed most of the concerns.
> We also appreciate the reviewer for further explaining the concerns and giving us the opportunity to address them.
>
> Firstly, we confirmed that the quantification method in ref [1] and our method are both applicable to cases with heterogeneous input and output. Thank you for pointing this out. Secondly, we fully understand the reviewer's concern about the potential similarity between our work and ref [1], and we provide our contributions compared to ref [1] below.
>
> ---
>
> The main contribution of our paper is to propose a definition, quantification, and utilization framework for heterogeneity in the MARL field.
> Under our framework, many previous works can be regarded as part of heterogeneity research, and therefore, it is inevitable that there will be some overlap with previous works. If only comparing ref [1] and our work, ref [1] can be seen as part of the starting point of our work, and we have comprehensively extended the work in ref [1].
>
>
> **A.** The research scope of ref [1] focuses on the policy diversity of agents; our research scope focuses on the heterogeneity of agents, providing a classification and definition of heterogeneity. In our work, we discuss five types of basic heterogeneity (including policy heterogeneity), and introduce the concept of meta-heterogeneity. Our definition of policy heterogeneity is compatible with the work in ref [1].
>
> **B.**  The quantification method in ref [1] is only applicable to quantifying policy heterogeneity and relies on explicit policy models. Our method frees from the model-based constraint and can quantify the heterogeneity without the need for models of POMG elements, making it suitable for quantifying more types of heterogeneity, as well as meta-transition heterogeneity (a virtual mapping constructed to quantify the comprehensive heterogeneity of agents, which can only be quantified in a model-free way; details in Appendix G).
>
> **C.**  The parameter sharing method in ref [1] (MADPS) utilizes the policy heterogeneity quantification and adjusts the parameter-sharing paradigm through a rule-based approach. This method requires task-specific hyperparameters such as division threshold, fusion threshold, and adjustment interval; our method (HetDPS) can utilize the quantification of various types of heterogeneity (meta-transition heterogeneity used in the experiment) and adjust the parameter-sharing paradigm through a two-stage grouping approach based on *distance-based clustering* and computing the *optimal bipartite graph matching* between two consecutive clusters. This approach avoids introducing hyperparameters such as fusion threshold and cluster number, and is insensitive to the adjustment interval hyperparameter (details in Supplementary Material Section-B.3).
>
> The table below provides a more intuitive and detailed comparison of our method with ref [1].
>
> | Compared Aspect | **Our Work** | Reference[1] |
> |--------|-----------|--------------|
> | Main Contribution | *Definition, quantification and utilization of Heterogeneity* | Quantification and utilization of policy diversity |
> | Types of Heterogeneity Involved | *observation heterogeneity, response transition heterogeneity, effect transition heterogeneity, objective heterogeneity, policy heterogeneity, meta-transition heterogeneity* | policy heterogeneity |
> | Quantification Method (Distance Calculation Part) | *Uses probability density function for auxiliary calculation; Uses agent-environment interaction data for calculation without full space traversal* | Does not use probability density function for calculation; Traverses full space of independent variables / subset of independent variable space |
> | Quantification Method (Representation Learning Part) | *Applicable to both model-based and model-free scenarios, can quantify various types of heterogeneity; Applicable to heterogeneous input-output situations* | Applicable to model-based scenarios (explicitly accessible policies), can only quantify policy heterogeneity; Applicable to heterogeneous input-output situations |
> | Heterogeneity Utilization Method (Parameter Sharing Part) | *Based-on quantification of various types of heterogeneity (meta-het in experiments); Distance-based clustering and bipartite graph matching dynamic grouping algorithm; Not depend on adjustment interval (details in supplementary material B.3)* | Based-on quantification  of policy heterogeneity; Rule-based dynamic grouping algorithm; Depends on division threshold, fusion threshold, and adjustment interval hyperparameters |
>
> ---
>
> We hope our responses above have fully addressed your concerns, effectively dispelling any initial reservations about this work. We extend our gratitude once more for your reply, and look forward to your feedback.

---

### Official Review · Reviewer_rRe8 · 2025-07-03

**Clarity:** 3
**Significance:** 3
**Originality:** 3
**Rating:** 4
**Confidence:** 4

**Summary:**

This paper introduces the idea of heterogeneity in multi-agent reinforcement learning.

They break this down into five types of heterogeneity: Observation heterogeneity, Response transition heterogeneity, effect transition heterogeneity, objective heterogeneity and policy heterogeneity. They then define a measure for quantifying the distances for heterogeneity, specifically in equation (3). To compute the heterogeneity distances they learn a representation mapping using a conditional variational autoencoder framework. They then use this encoders output for each sample gathered by the agents to calculate the heterogeneity distances for the various types.

They then have a case study which is multi-agent spread. A scenario in which agents are to move close to randomly generated landmarks which match their colour assignment. They do this with 6 extended versions to show the quantitative results of the different types of heterogeneity.

Ultimately this case study shows that fully parameter sharing has low policy-heterogeneity (and lower reward) and no parameter sharing has high policy-heterogeneity (higher reward). This reveals policy evolution differences.
The Meta-heterogeneity stays consistent, which reveals environmental characteristics.

They then develop Heterogeneity-based multi-agent Dynamic Parameter Sharing (HetDPS). This method adapatively groups agents for parameter sharing as the policy learns based on the heterogeneity types. They show its results in Particle based Multi-agent Spreading, as well as in the StarCraft Multi-Agent Challenge (SMAC). They compare to multiple baselines outlined in table 3. Showing that they perform well in 4 of the SMAC environments compared to the other methods, with comparable results to full parameter sharing in the environments that have homogenous agents.

Overall, I believe this is promising work, but needs work to help with clarity of the final contributions.

**Questions:**

1. Will the methods described to find heterogeneity of the different types help to identify lazy agents and help them be reassigned?
2. Do you think these methods to find heterogeneity can help agents match conventions in a setting strict coordination via one of multiple symmetrical but different conventions is required? (for example, different countries have different rules for driving on different sides of the roads.)

**Ethical Concerns:**

["NO or VERY MINOR ethics concerns only"]

**Final Justification:**

In line with the latest additions and edits that the authors have given, I am upping the score by 1 point for possible acceptance.

Opinion in support of this work:
I do believe there is merit to this work, and the ideas of heterogeneity proposed here does have potential to help future MARL research. I believe that diversity/heterogeneity is still an under-explored concept in MARL work. This work positively adds to MARL research.

**Limitations:**

Yes.

**Paper Formatting Concerns:**

Nothing major.

**Quality:**

2

**Strengths And Weaknesses:**

Strengths:

1.	Outlines and defines the heterogeneity types well.

2.	Outlines a method to get the heterogeneity distances. It is shown to work in the case study.

3.	Shows the method works on multiple domains and compares to multiple other methods. With it being better in quite a number of these domains.

Weaknesses:

1.	The paper is quite difficult to follow with most of the relevant information being in the appendix. Especially for HetDPS.

2.	The results are difficult to interpret, and could definitely use some more explanation. I’m not entirely sure what the Meta-Het Matrix is showing in Figure 6 for example, I assume it’s the different units from HetDPS’s measurement? If so, can it be made clear which unit type it is, so the heterogeneity is more obvious.

3.	Some of the graphs, specifically Figures 8 and 9, aren’t very clear as to what is to be interpreted.

---

> ### Author Rebuttal · Authors · 2025-07-31
>
> Thanks for your comment!
>
> We thank Reviewer rRe8 for considering *our work promising*, acknowledging that *our classification and definition of heterogeneity are well-established*, *appreciating the specific quantification method for heterogeneous distance*, and *the methodology for heterogeneity utilization that works* across multiple domains over other methods.
>
> In the following sections, we will provide our responses to the potential weaknesses pointed out by the reviewer and answer the two interesting questions raised.
>
>  ---
>
> **R2-1** *The information in the appendix is difficult to follow*
>
> **We notice that most of the weaknesses mentioned by the reviewer revolve around the appendix of the paper.**
>
> First, we acknowledge that the content in this paper, especially in the appendix, is indeed uncommon in conventional MARL research and somewhat abstract. This is because **the main contribution of this paper is to integrate approaches to heterogeneity from multiple disciplines, incorporate existing MARL research, and ultimately propose a practical methodology applicable in the MARL domain.**
>  Therefore, when writing the appendix, we aimed to provide readers with as much background knowledge and our thoughts as possible, **allowing readers to selectively read based on their own backgrounds.**
>
> We appreciate the suggestions regarding the HetDPS introduction, and **we have rewritten the sections in the appendix.** Moreover, **we have provided reproducible source code in the supplementary materials, which includes detailed readme files and code annotations, hoping to help readers understand the proposed method.**
>
> **R2-2** *More explanation about experimental results*
>
> The Meta-Het Matrix mentioned by the reviewer in Figure-6 was designed to demonstrate that our proposed HetDPS algorithm **has strong interpretability**. The four tasks in Figure-6 correspond to two scenario groups (*3s5z*&*3s5z_3s6z*, *MMM*&*MMM2*), where the agents' basic physical properties in each group are identical, yet the quantified heterogeneous distances differ significantly.
> **This confirms our conclusion that heterogeneity is not only related to the agents' own attributes, but also to their interactions with the environment.**
>
> Additionally, **we have uploaded more experimental details and result analyses in the supplementary materials. We have also moved this content to the main text.**
>
> **R3-3** *Some figures in the appendix are difficult to understand*
>
> There is no Figure 9 in our paper; we assume the reviewer is referring to Figures 7 and 8. These two figures in the appendix are used to explain MDP-based modeling related to RL and MARL. It is important to emphasize that **these two figures are cited from other papers, not our own**. As mentioned in R2-1, we aim to provide as much background knowledge as possible to help readers selectively understand. We used these two figures to explain why we chose POMG as the primary problem for discussing heterogeneity in MARL.
>
> ---
>
> **Q2-1**
>
> **A**:  This is a very intriguing question. We believe the answer is *yes*. We consider that the method of reassignment through grouping remains feasible. However, the foundation for grouping would not be the heterogeneity distance between agents, but rather treating the environment itself as a virtual agent and quantifying the transition-related heterogeneity distance between the *virtual  environment agent* and specific agents. Agents with large distances can be viewed as a certain degree of lazy-agents.
>
> We appreciate your inspiration, and combining our heterogeneity methodology with lazy agents in the MARL domain is a promising research direction.
>
> **Q2-2**
>
> **A**:
> We thank the reviewer for highlighting this important application. Our heterogeneity-quantification method can serve as a core tool to help match conventions and can assist as a constraint condition during policy training or deployment phases.
>
> The reviewer might be concerned that certain unconventional situations, such as symmetry, could potentially compromise the method's effectiveness. In fact, the features corresponding to our method do not necessarily need to be conventional features, and can be transformed according to specific scenarios. For instance, although the optimal properties of cars in left-hand and right-hand driving lanes might temporarily appear heterogeneous, selecting features unrelated to symmetry can ensure that optimal policies remain consistent (homogeneous) within our method, thereby establishing a constraint.
>
> ---
>
> Finally, we sincerely appreciate the reviewer's time and effort during the review process. We hope that the above responses thoroughly address your concerns and provide a basis for a more comprehensive evaluation of our work. If you have any additional thoughts or suggestions, we remain open to further communication. We look forward to your valuable feedback.

---

> > ### Comment · Reviewer_rRe8 · 2025-08-05
> >
> > R2-1
> > Rewriting and moving more information to the main paper will help a lot.
> >
> > R2-2
> > Perhaps I am interpreting the statement below incorrectly:
> > "where the agents' basic physical properties in each group are identical"
> > Lets take 3s5z as an example. How do stalkers and zealots have identical physical properties? They are very different.
> > Your meta-het matrix seems to show this quite well, which is a good sign.
> > The MMM and MMM2 matrices seem to show similarity between similar units.
> > It is difficult to interpret which agent index is which unit since we only have numbers and no mapping to the unit type, and there may be incorrect conclusions made based on this.
> > I will assume the mapping is 0-2 is stalkers and 3-7 are zealots. The meta-het values seem to show this, but I can't be sure without the mappings.
> > This however does not seem to be consistent from in the grids for 3s5z to 3s5z_vs3s6z. Unless 1 of the stalkers is behaving completely different to the others in 3s5z, and 2 of the zealots are completely different as well.
> > Please can you double check these mappings, and add some description to make these mappings clear.
> > Clearly showing these mappings might help interoperability.
> >
> > R3-3
> > To clarify, I type with my numpad, I accidentally pressed 8 and 9 instead of 5 and 6, I was still referring to those results as it is not immediately clear which agent is which, and could possibly lead to incorrect conclusions.
> > This was just reiterating the previous weakness.
> >
> > Thanks for the answers on the questions. I do believe there is an interesting direction for future research with your work.
> >
> > Overall, I believe this is good work, however taking into account the other reviews as well as my own, the paper could use some fine tuning. I will keep my score.
> > I do hope to see a polished version and for it published soon.

---

> > > ### Author Response · Authors · 2025-08-05
> > >
> > > Dear Reviewer rRe8,
> > >
> > > We sincerely appreciate your valuable comments and detailed explanations.
> > >
> > > ---
> > >
> > > *Response to the first-issue (R2-2)*:
> > >
> > > As you correctly pointed out, our statement in the rebuttal phase was not rigorous enough. What we intended to clarify was that the basic physical properties "*between different groups of agents* (across tasks) are identical", rather than "*among agents within the same group* (within a task) are identical". For example, in both the *MMM* and *MMM2* tasks, the MARL-controlled agents consistently include 2 Marauders, 7 Marines, and 1 Medivac unit. Their corresponding agent compositions are identical, but the physical properties of agents within the same group may differ.
> > >
> > > As suggested by the reviewer, we provide the relationship between unit-type and ID of agents in SMAC-related experiments:
> > >
> > > *Distribution of agent types in SMAC task (sorted in ascending order of ID)*
> > >
> > > | Task         | Agent Type Distribution                     |
> > > |--------------|---------------------------------------------|
> > > | *3s5z*         | **3 Stalkers** (0\~2) – **5 Zealots** (3\~7)          |
> > > | *3s5z_vs_3s6z* | **3 Stalkers** (0\~2) – **5 Zealots** (3\~7)          |
> > > | *MMM*          | **2 Marauders** (0\~1) – **7 Marines** (2\~8) – 1 **Medivac** (9) |
> > > | *MMM2*           | **2 Marauders** (0\~1) – **7 Marines** (2\~8) – 1 **Medivac** (9) |
> > >
> > >
> > >
> > > In our experiments based on Particle-based Multi-agent Spreading, our method effectively distinguished different agent categories. The obtained Meta-Het matrix perfectly corresponds with the agent type distribution, which intuitively makes sense.
> > >
> > > However, in more complex SMAC scenarios, we observed phenomena that were less aligned with intuition. Specifically, we found that in two super-hard difficulty tasks (*3s5z_vs_3s6z* and *MMM2*), the Meta-Het distance matrix showed strong consistency with agent type distribution. In contrast, for the easier tasks (*3s5z* and *MMM*), agents tended to evolve distinct grouping patterns. This discrepancy might originate from the requirement of maintaining specialized division of labor in hard tasks, which is less critical in easy tasks.
> > >
> > > Regardless, comparative analysis between *3s5z* and *3s5z_vs_3s6z* (or *MMM* and *MMM2*) corroborates the statement in the abstract and Section-3: Heterogeneity in MARL arises not only from agents' inherent functional differences, but also from agent-environment interactions. Furthermore, this analytical process demonstrates the strong interpretability of our proposed HetDPS algorithm.
> > >
> > >
> > > Thank you for your suggestion. We have added the above table to Section-6.2 of the main text.
> > >
> > > ---
> > >
> > >
> > > *Response to the second-issue (R2-3)*:
> > >
> > > Thank you for further explaining your previous concerns. We apologize for uploading the incorrect versions of Figure-5 and Figure-6 in our initial submission. The correct versions should be Figure-1 and Figure-2 in the supplementary materials, which are clearer and more aesthetically pleasing compared to the old figures. We have replaced the figures.
> > >
> > > ---
> > >
> > >
> > > We hope that the above response addresses your concerns. Once again, thank you for your comment, which is very helpful in improving the clarity of our work. We look forward to your feedback.

---

> > > > ### Comment · Reviewer_rRe8 · 2025-08-05
> > > >
> > > > This does help the clarity a lot.
> > > > Thank you.
> > > > I edited the score.
> > > >
> > > > In line with the latest additions and edits that the authors have given, I am upping the score by 1 point for possible acceptance.
> > > >
> > > > Opinion in support of this work: I do believe there is merit to this work, and the ideas of heterogeneity proposed here does have potential to help future MARL research. I believe that diversity/heterogeneity is still an under-explored concept in MARL work. This work positively adds to MARL research.

---

> > > > > ### Author Response · Authors · 2025-08-06
> > > > >
> > > > > Dear Reviewer rRe8,
> > > > >
> > > > > Thank you again for your feedback and further positive affirmation of our work. We appreciate your constructive suggestions, which help improve the clarity of our research.
> > > > >
> > > > > Sincerely, The Authors

---

### Official Review · Reviewer_9fzv · 2025-07-03

**Clarity:** 2
**Significance:** 2
**Originality:** 3
**Rating:** 4
**Confidence:** 4

**Summary:**

This paper focuses on different aspects of MARL related to homogeneity, by defining this notion for the multiple components of MARL, by proposing a way to measure such homogeneity and, finally, by leveraging this measure for parameter sharing. The performance of the proposed approach is evaluated both in a particle-based environment and in SMAC.

**Questions:**

- Could the authors be more specific on how the measure of heterogeneity is used in parameter sharing?
- HetDPS is claimed to achieve "optimal or comparable results", but that does not seem to be the case in 15a_3c (Figure 5), where HetDPS seems to alternate between optimal and sub-optimal behaviour between seeds. In what sense is this behaviour optimal?

**Ethical Concerns:**

["NO or VERY MINOR ethics concerns only"]

**Final Justification:**

The authors have addressed some of my concerns, so I increase my score.

**Limitations:**

Yes

**Paper Formatting Concerns:**

No concerns

**Quality:**

2

**Strengths And Weaknesses:**

Strengths:
- The proposed approach is motivated in a formal way
- The figures are well-designed and informative

Weaknesses:
- The definitions in Section 3 are highly redundant and straightforward (except perhaps for Definition 3), this space could have been used for more insightful aspects of the work
- Proposition 1 is very similar to existing results in the literature, see e.g. Theorem 3.3 in [R1], which shows the same result for Wasserstein distances (which is also the distance used in the paper)
- The proposed method should be introduced in more details

[R1] Lin, Z., & Ruszczyński, A. (2023). An integrated transportation distance between kernels and approximate dynamic risk evaluation in Markov systems. SIAM Journal on Control and Optimization, 61(6), 3559-3583.

---

> ### Author Rebuttal · Authors · 2025-07-31
>
> Thanks for your comment!
>
> We appreciate Reviewer 9fzv find our work is motivated in a formal way, the figures are well-designed and informative. The concerns raised by the reviewer primarily stem from misunderstandings about our work. We will clarify these points one by one and address the related questions in hopes of achieving a constructive discussion with the reviewer.
>
>  ---
>
> **R1-1** *Regarding the Redundancy and Straightforwardness of Definitions in Section-3*
>
> **The definitions in Section-3 are fundamentally aimed at categorizing and subsequently quantifying heterogeneity.** As stated in our paper, heterogeneity in MARL can be classified in multiple ways (we selected the most suitable way for quantification and utilization), which can also lead to various definition approaches. These classifications and definition methods originate from different perspectives,  and should not be judged as right or wrong. (We discussed more potential heterogeneity in the MARL domain in Appendix D)
> Our definition approach is designed to clarify which elements of POMG are associated with each type of heterogeneity, thereby laying the groundwork for subsequent quantification.
>
> On the other hand, the reviewer might consider the conditions for judging or identifying heterogeneity based on the definition too simple, as we treat any situation with non-complete homogeneous as heterogeneous.
>
> **However, the straightforwardness of the definition should not be considered a weakness of the paper, as our work goes beyond the definition itself, and our method can further identify and quantify heterogeneity.**
>
>
> **R1-2** *Comparison of Proposition-1 with Reference [R1]*
>
> Reference [R1] proposed a distance method between kernels, which also utilized metrics with distance metrics like WD as a foundation. **To clarify, we did not aim to prove the properties of WD, but rather to leverage its properties to demonstrate the characteristics of our proposed method.** We proved the properties of the proposed strict heterogeneity distance in the appendix, and discussed the properties of the heterogeneity distance calculated via practical methods. **Moreover, Proposition-1 represents the properties of the heterogeneity distance we proposed, not our innovation.**
>
> **R1-3** *Introducing More Method Details*
>
> We must acknowledge that due to the comprehensive content of the work and the submission length constraints, we could not exhaustively include every methodological details in the main text. *Therefore, we placed more foundational conceptual content and method details in the appendix.*  **In the main text, we retained the core idea of the method and its connection to the overall narrative.**
>
> **On the other hand, we also provided reproducible source code in the supplementary materials, which includes detailed readme files and code annotations, hoping to help readers understand the proposed method.**
>
> We appreciate the suggestions regarding our paper's structure in the review, and we have moved some details about HetDPS from the appendix to Section-5.
>
> ---
>
>
> **Q1-1** *Relationship between Heterogeneity Quantification and Dynamic Parameter Sharing*
>
> **A**:  We developed a dynamic parameter sharing algorithm that, quantifies Certain Heterogeneity Distance between agents (Meta-Het in our experiments) at regular intervals, and adjusts parameter sharing among agents based on this distance.
>
> Specifically, during the quantification cycle, the algorithm first calculates the heterogeneity distance matrix of agents, and then groups agents using a distance-based clustering method (Affinity Propagation in our experiments). However, considering that the grouping results might differ between consecutive cycles. Directly allocating policies based on the current grouping could disrupt existing policy inheritance and potentially cause policy training collapse. Therefore, we modeled the policy grouping between consecutive cycles as a bipartite graph matching problem, carefully calculating the optimal matching through case-by-case analysis to ensure each agent is assigned the most suitable policy network.
> Consequently, the method is insensitive to algorithmic cycle hyperparameters due to its ability to "smoothly" find the optimal agent policy allocation. Moreover, by periodically quantifying agent heterogeneity, the algorithm provides sufficient interpretability for evolutionary analysis.
>
> **Q1-2** *Explanation of Behaviour Optimal*
>
> **A**:
> We apologize for uploading an incorrect version of Figures 5/6. We have uploaded corrected versions of these figures in the supplementary materials, specifically in Section-B's Figures 1/2. From the supplementary Figure-1, it is evident that HetDPS achieves optimal performance across all Particle-based Multi-agent Spreading tasks (including the *15a_3c* task mentioned by the reviewer). We have replaced the original Figures 5/6 with these updated versions.
>
> ---
>
> Finally, we sincerely appreciate the reviewer's time and effort during the review process. We hope that the above responses address your concerns. If you have any thoughts or suggestions, we welcome further communication! We look forward to your reply.

---

> > ### Comment · Reviewer_9fzv · 2025-08-04
> >
> > R1-1: The notions can be kept, but it would have been enough (at least for Definitions 1,2,4,5) to define homogeneity as equality in terms of conditional distributions of one of the elements in the POMG, and talk, e.g., about $\Omega$-homogeneity or $\mathcal{T}$-homogeneity, which would have been clear enough and would have saved half a page of definitions. In particular, this would have allowed to include more details about the proposed method (see R1-3).
> >
> > R1-2: The usual understanding is that if a proof is provided then the result is new (otherwise something like "the proof is included for completeness" is usually added). In any case, a reference to the literature should be included.

---

> ### Author Response · Authors · 2025-08-04
>
> Dear Reviewer 9fzv,
>
> We sincerely appreciate your valuable comments.
>
> ---
>
> *Response to Comment-1*:
>
> Thank you for your constructive suggestion. Following your recommendation, we have streamlined the description in the definition part and consolidated its content into a single table. This optimization successfully saved 11 lines of space while enhancing content presentation. The newly available space enabled us to incorporate essential content from Appendices G and I into the main text, and to provide additional explanations regarding our proposed quantification methodology and implementation approaches.
>
> Specifically, the revised table is structured as follows. We have simplified certain symbolic representations, where $\neg$ and $\land$ correspond to the logical operators "negation" and "conjunction" respectively.
>
> | Heterogeneity Type | Heterogeneity Description | Related POMG Elements | Mathematical Definition |
> |---|---|---|---|
> | Observation Heterogeneity | Describes the differences of agents in observing global information | Agent's observation space and **observation function** | $\neg (O^i = O^j \land \forall \hat{s} \in \lbrace S^i \rbrace_{i \in N}, \Omega^i(\cdot\|\hat{s}) = \Omega^j(\cdot\|\hat{s}))$ |
> | Response Transition Heterogeneity | Describes the differences of agents in how their state transitions are affected by global environment (*environment-to-self*) | Agent's state space and **local state transition function** | $\neg (S^i = S^j \land \forall \hat{s} \in \lbrace S^i \rbrace_{i \in N}, \hat{a} \in \lbrace A^i \rbrace_{i \in N}, \mathcal{T}^i(\cdot\|\hat{s}, \hat{a}) = \mathcal{T}^j(\cdot\|\hat{s}, \hat{a}))$ |
> | Effect Transition Heterogeneity | Describes the differences of agents in how their states and actions impact global state transitions (*self-to-environment*) | Agent's action space, state space, and **global state transition function** | $\neg (S^i = S^j \land A^i = A^j \land \forall s' \in S^{-i}, a' \in A^{-i}, s \in S^i, a \in A^i, \mathcal{T}^{-i}(\cdot\|s', s, a', a) = \mathcal{T}^{-j}(\cdot\|s', s, a', a))$ |
> | Objective Heterogeneity | Describes the differences of agents in the objective they aim to achieve | Agent's **reward function** | $\neg (\forall \hat{s} \in \lbrace S^i \rbrace_{i \in N}, \hat{a} \in \lbrace A^i \rbrace_{i \in N}, r^i(\cdot\|\hat{s}, \hat{a}) = r^j(\cdot\|\hat{s}, \hat{a}))$ |
> | Policy Heterogeneity | Describes the differences of agents in their autonomous decision-making based on observations | Agent's observation space, action space, and **policy** | $\neg (O^i = O^j \land A^i = A^j \land \forall o \in O^i, \pi_i(\cdot\|o) = \pi_j(\cdot\|o))$ |
>
> ---
>
> *Response to Comment-2* :
>
> We appreciate your clarification of previous concerns.
>
> While there are conceptual similarities between the proof framework in *[R1]* and ours, our heterogeneity-distance calculation incorporates *probability density functions* of variables. Direct application of their proof methodology without modification may be inappropriate. Moreover, our further discussion extends to heterogeneity-distance formulations involving representative standard distribution, which significantly affects the proof of the property *Identicals of indiscernibility*.
>
> As you correctly pointed out, we must explicitly identify the “new” aspects of our proof while properly citing related literature. To address this, we have included a clear citation of *[R1]* in the method section and the related work section, and explicitly articulate the methodological differences between our approach and existing works.
>
> ---
>
> In summary, we are grateful for your insightful comments that substantially improved the clarity of our work. We look forward to your feedback.

---

### Note · Authors · 2025-08-12

**Dear PC/SAC/AC/Reviewers**,

We would like to express our sincere gratitude to all the reviewers and AC for their valuable contributions and efforts, which have facilitated a constructive discussion phase.

---

We appreciate that all the reviewers found **our research on heterogeneity is in-depth and formal**. We are also encouraged by the reviewers' appreciation that **our proposed heterogeneity quantification method is insightful (dH7A) and effective (rRe8, 21mt, dH7A)**, that **our practical algorithm for utilizing heterogeneity is commendable (rRe8, 21mt, dH7A) and has strong empirical results (rRe8, dH7A)**, that **our overall framework for addressing heterogeneity has broken through the limitations of previous work (rRe8, 21mt) and has a promising impact on MARL future research (rRe8, dH7A)**. Additionally, our paper has been praised for being clear and well-organized (dH7A), and the figures are deemed well-designed and informative (9fzv).

---

In addition to the reviewers' positive comments, we appreciate the useful suggestions they provided. Based on their suggestions, we have made the following fine-tuning to our paper:

1. We have condensed the definition section into a table (in response to 9fzv).

2. We have moved details of methodology from the appendix to the main text and enriched the experimental section (in response to 9fzv, rRe8).

  3. We have provided a more explicit comparison with related works (in response to 9fzv, 21mt).

**Although these refinements are relatively minor, they have significantly improved the clarity of our paper.**

---

In addition to the paper revisions, **we have also provided responses and experiments to thoroughly address the reviewers' concerns**:

1. We provided a detailed comparison with 3 related works. Our proposed framework is fully compatible with their work. Our method differ fundamentally from theirs, addressing their pain points (e.g., inability to model-free scenarios and dependence on task-specific hyperparameters) (in response to 21mt).

2. We tested our method under different distance metrics (BD, WD, HD), demonstrating generality (in response to 21mt).

3. We conducted comparisons and scaling experiments with different quantification methods (1-WD, FR), showing our method's robustness to agent scaling and accurate group identification (in response to 21mt, dH7A).

---

Finally, we would like to thank all the reviewers and AC once again for their time and effort.

**Sincerely, The Authors.**

---

### Decision · Program_Chairs · 2025-09-17

**Decision:**

Reject

**Comment:**

This paper is right at the threshold between acceptance and rejection. It definitely has its worthwhile contribution: all reviewers praised the interesting novel categorization of different types of heterogeneity in MARL. However, the following theory appears to be (at least in parts) quite similar to existing literature, some of the authors claims can be misunderstood easily, and the application of the theory to parameter sharing are incremental at best. Some reviewers also found the results hard to interpret and were missing clearer evidence of the hypotheses. The SMAC evaluation misses some default MARL baselines (e.g. QMIX). To their credit, the authors added some missing ablations during the rebuttal.

Many reviewers recommended nonetheless acceptance, as they valued the novel categorization framework. For example dH7A noted: "Although the performance gains are not dramatic, the conceptual novelty, rigorous framework, and practical insights outweigh these weaknesses. I think this work will help structure future MARL research and serves as a strong foundation for follow-up studies."

On the other side, the paper does not contribute a lot beyond this categorization. For example, 21mt criticized: "The main issue with this paper is not insufficient experiments but its similarity to existing work [1], [..] this work's primary contribution is its extended definition of diversity compared to [1], as the authors propose more types of heterogeneity. However, the core qualification methods of the two approaches are quite similar, from their network structures to their loss functions."

I ended up siding with the argument to reject the paper, but I do acknowledge that it does have its value and would not be unhappy if it gets bumped up.